

# Global synthesis of forest cover effects on long-term water balance partitioning in large basins

Daniel Mercado-Bettín[a,1], Juan F. Salazar[a], and Juan Camilo Villegas[a]

[a]Grupo GIGA, Escuela Ambiental, Facultad de Ingeniería. Universidad de Antioquia. Medellín, Colombia

*Correspondence to:* Daniel Mercado-Bettín (daniel.mercado@udea.edu.co)

**Abstract.** Global changes in forest cover have been related to major scientific and social challenges. There are important uncertainties about the potential effects of ongoing forest loss on continental water balances. Here we present an observation-based analysis of long-term water balance partitioning (precipitation divided into evaporation and runoff) in 22 large basins of the world, whereby we identify two partitioning patterns likely related to biophysical mechanisms that depend on the presence and abundance of forests. In less forested basins, evaporation dominates water balance and, as forest cover increases, this dominance of evaporation over runoff is reduced. When forest is the predominant cover, both components account for nearly half of precipitation in the long-term water balance. The distinction between these two patterns is not fully explained by differences between water- and energy-limited environments, but requires consideration of other biophysical properties that affect precipitation and its conversion into evaporation and runoff. Our results indicate that forest cover is an effective descriptor of basin attributes that are relevant for characterizing long-term water balance partitioning in large basins of the world. Further, our results provide insights to understanding and predicting the potential consequences of forest loss on continental water availability, a critical determinant for multiple ecological and societal processes.

## 1 Introduction

A major scientific question in hydrological sciences is how river flows (and therefore water availability for multiple social and ecological processes) will change in response to changes in forest cover, either forest loss or forest encroachment (Zhang et al., 2016; Ellison et al., 2012; Zhou et al., 2015). Two contrasting views have been presented for answering this question (Ellison et al., 2012). One view is that the presence of forests causes a decrease in river flows, mainly because forests can support large evaporation fluxes (which includes free surface evaporation and plant transpiration) due to their large cumulative leaf area. A contrasting (and less traditional) view is that the presence of forests can lead to an increase of river flows through, for instance, complex land-atmosphere interactions related to feedbacks of vegetation on precipitation (e.g. precipitation recycling). Both views are supported by observational and modelling studies (Ellison et al., 2012; Zhang et al., 2016). For instance, previous studies have reported that forest cover reduction in large basins can result in both increased (Wei and Zhang, 2010) or decreased (Coe et al., 2009) mean river flows. Such contradictory views highlight that there is not a single, globally-applicable response to the initial question. Progressing towards quantitative understanding of the hydrological role of forests is a fundamental step



for predicting river flow regimes in a changing environment, especially under the perspective of the "*Panta Rhei*—Everything Flows" debate (Montanari et al., 2013).

One key difficulty in addressing questions about the hydrological and meteorological role of forests in basins arises from scale issues (D'Almeida et al., 2007; Zhang et al., 2016). Of particular importance is that results from small basins (e.g.

paired catchment studies) cannot be directly extrapolated to large basins. This because there may be complex land-atmosphere interactions that are not observable at the small scale but can have important implications for the potential effects of forest cover change on river flows at larger scales (e.g. Stickler et al., 2013; Coe et al., 2009). Precipitation recycling is an important example of such interactions. Global estimates indicate that, on average, 40% of the terrestrial precipitation originates from land evaporation and that 57% of all terrestrial evaporation returns as precipitation over land (Van der Ent et al., 2010). In the

Amazon, the largest basin of the world, a large fraction (estimates vary around $\sim 40\%$) of precipitation is recycled (Eltahir and Bras, 1994), i.e. a large fraction of the precipitation falling over the Amazon river basin has been originated in forests within the same basin as an evaporation flux. This and other related phenomena (e.g. production of biogenic cloud condensation nuclei, Pöschl et al. (2010); activation of shallow convection through transpiration, Wright et al. (2017)) establish a physical linkage between the presence of forests and the behavior of precipitation over the basin. Under this perspective, precipitation in a large

basin is not independent of forest cover (they are linked through observable biophysical mechanisms), and evaporation should not simply be assumed as a loss for the surface water balance, but rather as a potential component of hydrological regulation mechanisms in the basin (Salazar et al., 2017).

Scale issues and related land-atmosphere interactions can have important practical implications. For instance, Coe et al. (2009) showed that, in large tributaries of the Amazon river basin, modeling results about the effects of deforestation on

river flows are contradictory depending on whether forest feedbacks on precipitation are considered or not. In particular, they found that simulated river flows are reduced as a consequence of deforestation (with important implications for hydropower generation) when forest feedbacks on precipitation are considered, but not otherwise. The interactive mechanisms that link precipitation and evaporation through continental moisture recycling patterns is importantly related to land cover, and plays an important role in the distribution of global water resources (Van der Ent et al., 2010).

The partitioning of the long-term water balance (precipitation divided into evaporation and runoff) can be affected by basin attributes that include not only properties that are relatively invariant (e.g. geological properties and river network topology), but also properties that are highly sensitive to global change at policy-relevant time scales (e.g. land cover). Identifying those factors that are both highly sensitive to global change and strongly influential on the partitioning is fundamental for predicting the hydrological effects of global change. Vegetation cover and vegetation-related processes meet these two conditions in

many basins of the world (Spera et al., 2016; Sterling et al., 2013; Coe et al., 2009; Piao et al., 2007). We choose to focus on forests because these ecosystems are highly threatened worldwide (Hansen et al., 2010, 2013; Malhi et al., 2014), while there are important uncertainties about the potential consequences of forest loss on continental water balances (e.g. Bonan, 2008; Ellison et al., 2012; Makarieva et al., 2013; Zhang et al., 2016), including the possibility of forest loss tipping points (Boers et al., 2017; Zemp et al., 2017; Khanna et al., 2017; Lawrence and Vandecar, 2015).



In the the long-term land water balance equation,

$$P = E + R, \tag{1}$$

precipitation, ($P$) is divided into runoff ($R$) and evaporation ($E$) fluxes, under the assumption that changes in the land water storage within the basin are negligible (tend to zero) in the long term (Manabe, 1969; Zhou et al., 2015). The widely-recognized

Budyko hypothesis defines limits for this partitioning based on the availability of water and energy (Budyko, 1974). The maximum possible actual evaporation ($E$) is limited by the potential evaporation ($E_p$), i.e. the available energy. Mass continuity implies that $E + R$ is also limited by the available water, $P$. However, the specific partitioning pattern in a river basin (the observed values of $E$ and $R$) depends not only on the availability of water ($P$) and energy ($E_p$), but also on the biophysical processes and basin attributes that exert controls on the production of $E$ and $R$. This implies that same water and energy

availability ($P$ and $E_p$) can exist in basins with different hydrological partitioning patterns ($E$ and $R$), and leads to the important question of how these patterns relate with relevant biophysical attributes of such basins.

By means of an observation-based analysis, here we characterize long-term water balance partitioning in 22 large basins of the world, and explore the potential linkage between observed partitioning patterns and the extent of forest cover in the basins. Our approach is in the spirit of linking patterns to processes (Sivapalan, 2005), and of using data-intensive science as a timely

and promising paradigm for advancing hydrological science (Peters-Lidard et al., 2017).

## 2  Data and methods

The partitioning of $P$ into $E$ and $R$ is summarized by the *runoff coefficient k* which quantifies the fraction of $P$ that is converted into $R$, so that $R = kP$ (Sherman, 1932). Using river flow records from 177 gauges distributed among 22 basins of the world (Fig. 1a, Supplementary Table S1), we computed the value of $k$ at each gauge as $k = R/P$ averaged for the period 2001–2012.

$R$ was computed as $R = Q/A$, where $Q$ is the long-term average river flow (data from national and international databases, Supplementary Table S2) and $A$ is the drainage area at each gauge. $A$ values were obtained through the best basin delineation generated in the hydrological modules of GRASS GIS (http://grass.osgeo.org/) based on Digital Elevation Models (DEMs) extracted from the GTOPO30 (DAAC, 2004) and SRTM (Jarvis et al., 2008) projects. $P$ was computed as the cumulative value at each gauge, using the Tropical Rainfall Measuring Mission (TRMM-3B42) (Huffman et al., 2007) for tropical basins

(Magdalena and Amazon), and the European Centre for Medium-Range Weather Forecasts (ECMWF) ERA-Interim reanalysis (Dee et al., 2011) for the rest of the basins. For the analysis, we also used potential evaporation ($E_p$) computed as the cumulative value at each gauge, using the Global Land Evaporation Amsterdam Model (GLEAM v3.0a, Martens et al. (2017); Miralles et al. (2011)). In the Mackenzie, Lena and Vitim basins, we used data for only 6 months per year (May to October) to include biologically active vegetation and river flows dominated by rainfall-runoff processes. The snow-melt equivalent was subtracted

from runoff in these three basins.

To provide a metric of forest cover that relates to the statistics of hydrological partitioning in each basin, and considering that vegetation cover is not a static attribute, we constructed a global land cover map (Fig. 1a) using the temporal mode (the





most frequent class) for each pixel in the 12-year (2001–2012) map series of MODIS-MCD12C1 (Friedl et al., 2010). Land cover classification was defined after the International Geosphere Biosphere Programme (IGBP) scheme, which divides global land cover into 16 classes. We further grouped them into five classes: (1) Forest, which includes evergreen and deciduous forest types; (2) Shrub-Grass-Savanna, that includes two types of shrub-lands (open and closed), two types of savannas (woody and not) and grasslands; (3) Urban-Crop, that includes croplands, urban zones and cropland/natural mosaics; (4) Water that includes open water areas, wetlands and snow; and (5) Desert that includes barren areas (no strictly defined desert was present in the basins in this study).

To explore potential linkages between water balance partitioning and forest cover we use a suite of statistical techniques including correlation analysis (using sing Pearson's, Spearman's and Kendall's correlation methods, Supplementary Table S3) and locally weighted polynomial fittings (LOESS).

## 3   Observed patterns of water balance partitioning

Long-term water balance partitioning (represented by $k$) and cumulative forest cover fraction vary along the river network of each basin (Fig. 1b). There is no universal pattern for the variation of $k$ upstream from the outlet of each basin (left to right along the $x$-axis of Figure 1b), consistent with the spatial variability of $P$ and heterogeneity of the biophysical processes and attributes that affect the production of both $E$ and $R$.

The studied basins differ widely in their environmental characteristics, including geographical location, climatic regimes, geological and geomorphological properties, and land cover types. However, an analysis of the whole set of basins reveals two distinctive patterns of the long-term water balance partitioning. Basins in Figure 2 are ordered, from left to right, by total forest cover fraction (green shade). Box-plots describe the spatial variability of $R$ (Fig. 2a), $P$ (Fig. 2b) and $k$ (Fig. 2c) within each basin. A LOESS fitting ($p < 0.05$, blue line in Fig. 2c) indicates that the mean value of $k$ varies with the forest cover fraction in a way that coincides with two different patterns of water balance partitioning (Equation 2 and Fig. 3): an $E$-*dominated* pattern ($k < 0.5$ and, therefore, $E > R$) in the less forested basins, and a $P$-*halved* pattern ($k \approx 0.5$ implying that $E \approx R \approx P/2$) in the more forested basins.

The partitioning patterns described here correspond to two out of three theoretically-possible patterns, depending on the value of $R/E$ ratio. Since $R = kP$, mass continuity (Equation 1) implies that $E = (1 - k)P$ with $0.0 \leq k \leq 1.0$ and, therefore,

$$\frac{R}{E} = \frac{k}{1-k} \begin{cases} < 1.0, & \text{if } 0.0 \leq k < 0.5 \ (E\text{-dominated}) \\ = 1.0, & \text{if } k = 0.5 \ (P\text{-halved}) \\ > 1.0, & \text{if } 0.5 < k \leq 1.0 \ (R\text{-dominated}), \end{cases} \qquad (2)$$

where $0.0 \leq k < 0.5$ indicates that the partitioning pattern is $E$-dominated, meaning that most of $P$ is converted into $E$ and $R < E$. The opposite occurs if $0.5 < k \leq 1.0$, i.e. the pattern is $R$-dominated and $R > E$. The only alternative to these patterns is a $P$-halved pattern in which $P$ is equally divided into $R$ and $E$ ($k = 0.5$). All of these patterns can occur in nature. Therefore,



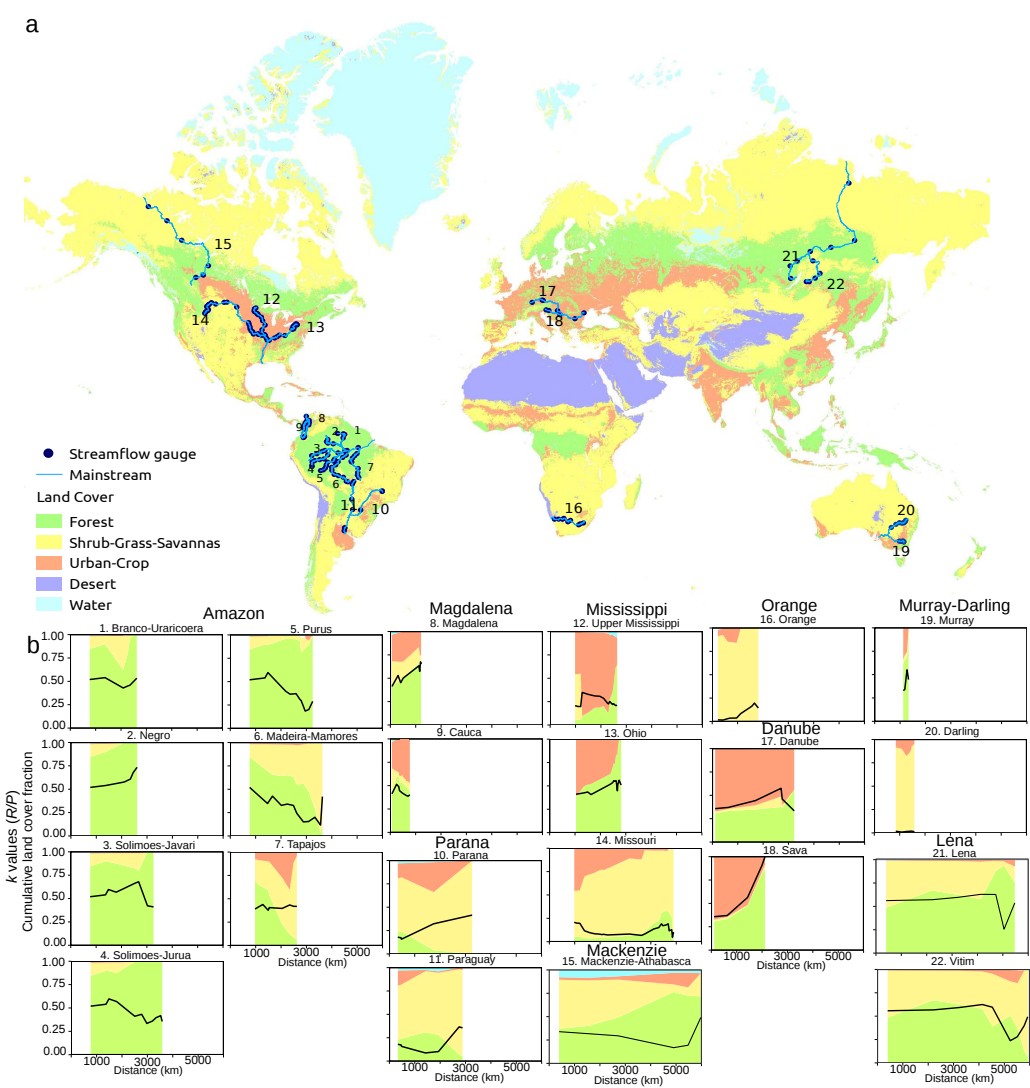

**Figure 1.** (a) Global basins selected for our analysis and associated global reclassified map of land cover mode (the most frequent class during 2001–2012). Numbers identify each basin for reference in b. (b) Cumulative fraction of land cover (spatial average) on each basin as a function of upstream distance to the basin outlet ($x$-axis). Colors represent the same categories as the map. Black lines correspond to $k$ values at each gauge along the river network of each basin.



**Figure 2.** Distribution of spatially averaged $R$ (a), $P$ (b) and $k$ (c) for the 22 basins organized by increasing forest cover fraction (green shade), for the 2001–2012 period. Boxplots describe the spatial variability of $R$ (a), $P$ (b) and $k$ (c) within each basin. In basins with low forest cover fraction, $k$-mean values (blue triangles) increase with forest cover fraction, with $k < 0.5$: $E$-dominated pattern. In basins with high forest cover fraction, $k$-mean values converge to a value around 0.5: $P$-halved pattern. Blue line is the LOESS fitting and grey shade is the corresponding 95% confidence interval.





long-term water balance partitioning in a given river basin can be schematically described by a point in the $xy$-space showed in Figure 3. Notably, the observed partitioning patterns in the studied basins are not characterized by $k$-values randomly distributed throughout this space. Instead, the observed $k$-values are organized in a way that coincides with the $E$-dominated pattern in the less forested basins, and the $P$-halved pattern in the more forested ones. The $R$-dominated pattern is not prevalent among the studied basins.

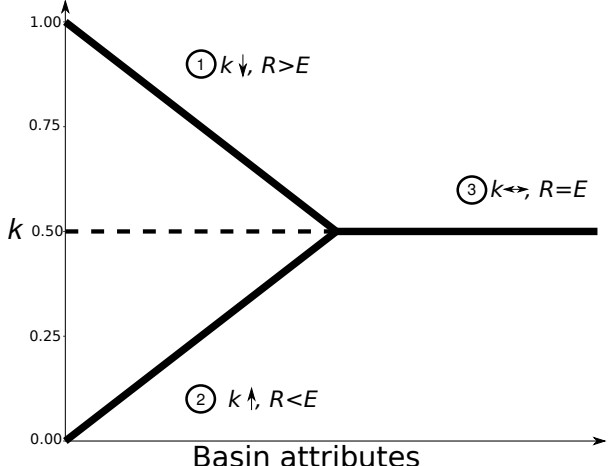

**Figure 3.** Conceptual patterns of long-term water balance partitioning that can occur in river basins. It can be (1) $R$-dominated ($k > 0.5$), (2) $E$-dominated ($k < 0.5$), or (3) $P$-halved ($k = 0.5$), depending on basin attributes schematically represented by the $x$-axis.

Notably, observed patterns of $k$ are not directly related to patterns in either $P$ nor $R$. These two variables, instead, exhibit different responses to forest cover. For instance, same $k$ values are found in basins with very different $P$ (e.g. Branco and Ohio), and similar $P$ values can result in very different $k$ values (e.g. Darling and Murray). This suggests that the variability of $k$ among river basins (the observed patterns), as well as its potential relation with forest cover, emerge from the conversion of $P$ into $R$, rather than being determined by $P$ input alone. A comparison between the Darling and Murray basins illustrates this observation. These basins are located in the same region (they are part of the same large basin), and receive a $P$-input that exhibits small variability and a similar mean value (Fig. 2b). However, the water balance partitioning in the Darling basin (the less forested one) is $E$-dominated, whereas it is $P$-halved in the Murray basin (the more forested one, Fig. 2c).

Another interesting comparison is that between the Missouri, Upper Mississippi and Ohio basins, that make up part of the same large basin of the Mississippi river. They are ordered (left-to-right in Fig. 2) by their mean values of $P$ and $k$, as well as by their forest cover fraction. Missouri and Upper Mississippi (the less forested basins) are $E$-dominated, while Ohio (the more forested basin) is $P$-halved. In this case not only $k$ but also $P$ and $R$ grow with increasing forest cover. Among these three basins, the maximum $k$ value (i.e. the maximum relative production of $R$) is close to 0.5 and occurs in the more forested basin: Ohio.



Overall, the studied basins can be divided into two different groups depending on their long-term partitioning pattern. Basins in the first group (from Orange to Vitim) are characterized by $k$ values that are generally lower than 0.5 (an $E$-dominated partitioning pattern), and forest cover fractions that are also lower than 0.5. Among these basins, we found a significant and positive correlation between $k$ and forest cover fraction ($\rho = 0.88$, $p < 0.001$, Supplementary Tables S4 and S5). This regression model was used to separate both groups of basins: it was fitted up until the point where correlation was maximized, corresponding to the Vitim basin. Basins in the second group (from Cauca to Negro) are characterized by $k$ values that are generally close to 0.5 (a $P$-halved partitioning pattern), and forest cover fractions that are higher than 0.5.

Independent of the potential mechanisms relating water balance partitioning and forest cover, the observed patterns challenge the view that the presence of forests reduces river flows (the "demand-side thinking" as described by Ellison et al. (2012)). Instead, our results are consistent with the contrasting view that the presence of forests enhances the long-term capacity of river basins to maintain larger river flows (the "supply-side thinking", Ellison et al. (2012)).

Artificial reservoirs and associated water resources schemes can affect the water balance partitioning ($k$). To explore this, we implemented the same methodology using 21 rivers (Supplementary Table S6) with an approximately natural flow and excluding basins where large amounts of water are transferred from the basin to external regions via water resources systems (e.g. the Murray, Darling and Orange basins). The LOESS fitting (Supplementary Figure S1) shows a similar pattern in the partitioning ($k$) as Figure 2c, associated with the conceptual patterns represented in Figure 3. The patterns found using these basins also show that: in basins with low forest cover fraction, $k$-mean values increase with forest cover fraction, with $k < 0.5$, following the $E$-dominated conceptual pattern; while, in basins with high forest cover fraction, $k$-mean values converge to a value around 0.5, following the $P$-halved conceptual pattern. The correlation between mean forest cover fraction and $k$-mean values using these basins (Supplementary Table S7-9) also support the above analysis. The resulting correlations using the 21 free-flowing river basins are mostly represented by the first 9 basins. The correlation using only the first 9 basins is higher and there are not significant correlation between the 11 remaining basins. In general, the results obtained using free-flowing river support the above patterns found and the statistics implemented using the 22 river basins.

A fundamental challenge in quantifying hydrological response (e.g. variations in the water balance partitioning) to forest cover change is to exclude the effect of non-forest drivers on runoff (Renner et al., 2014). This can be even more challenging for large basins with various confounding factors including artificial reservoirs and associated water resources schemes (Zhang et al., 2016). Although more-detailed studies are essential to understand water balance partitioning dynamics in different basins, as well as to characterize the influence of forest and non-forest drivers, our observation-based analysis allows to infer that variations in water balance partitioning patterns are related to variations in forest cover. Such empirical approaches are essential because it is becoming clear that accurate mechanistic models to predict hydrological response to forest cover change at multiple spatial and temporal scales are currently beyond our reach (Zhang et al., 2016), and predicting this response remains a fundamental challenge in environmental science today (Ellison et al., 2012; Montanari et al., 2013; Zhang et al., 2016).

Observed relations between partitioning patterns ($k$-values) and forest cover are intended to be only descriptive and not predictive. Observed differences between partitioning patterns in more or less forested basins cannot be directly attributed to the effect of forests on the long-term water balance partitioning in large basins, as correlation does not necessarily imply





causation. However, a growing body of scientific literature relates forest cover changes (e.g. deforestation) with alterations in river flow regimes (e.g. Sterling et al., 2013; Stickler et al., 2013; Zhou et al., 2015; Zhang et al., 2016), thereby implying that statistical correlations between river flow- and forest cover-related variables are not necessarily spurious, but rather can be a consequence of forest-related biophysical mechanisms. This is consistent with the general idea that, due to the potential effects

of many confounding factors that can affect runoff in large basins, and the associated uncertainty of any method, we can only draw statistical inference about the hydrological effects of forests (Zhang et al., 2016). In the next section we will discuss the potential linkage between the observed partitioning patterns and the presence of forests in large basins.

## 4   Discussion

### 4.1   Water- and energy-limited environments

The Budyko hypothesis allows to classify hydrological systems, including, river basins as water- or energy-limited, depending on whether the ratio between potential evaporation ($E_p$ representing available energy) and precipitation ($P$ representing available water) is higher or lower than 1, respectively. In a Budyko context, observed patterns of water balance partitioning ($k$) are not directly the result of patterns in either water availability ($P$, Fig. 2b) nor energy availability ($E_p$, Fig. 4a). Same $P$-values can be associated with different partitioning patterns (e.g. Murray and Darling), and same partitioning patterns can

be found in basins with different $P$-values (e.g. Ohio and Branco). Indeed, the $P$-halved pattern is common to basins where $P$ varies from less than 1,000 mm/year to more than 2,000 mm/yr (Fig. 2). Similarly, differences in $E_p$ between less-forested and more-forested basins (Fig. 4a) do not coincide with the distinction between $E$-dominated and $P$-halved patterns (Fig. 4c). Same values of $E_p$ can be associated with different partitioning patterns (e.g. Missouri and Ohio).

The $E_p/P$ ratio (Fig. 4b) and the partitioning pattern ($k$, Fig. 4c) are not independent because they both depend on $P$. Less

forested basins, where the partitioning pattern is $E$-dominated, are generally closer to water-limited environments ($E_p/P > 1$); while the more forested basins, where the partitioning pattern is $P$-halved, are more concentrated in the region of energy-limited environments ($E_p/P < 1$). However, variations in $E_p/P$ do not entirely coincide with the observed partitioning patterns. The $E$-dominated pattern does not only occur in water-limited, less-forested, basins (exceptions include Parana, Paraguay and Upper Mississippi where $E_p/P \leq 1$), and the $P$-halved pattern is not exclusive of energy-limited, more-forested, basins (e.g.

Murray is not energy-limited but its partitioning pattern is $P$-halved).

Most (but not all) of the more forested basins are energy-limited environments (Fig. 4b). This implies that there is an excess of water in the surface that could be transformed into runoff, likely leading to an $R$-dominated pattern. However, the $R$-dominated pattern is not prevalent in the more forested basins —few exceptions include some gauges in the Sava river where $k$ reaches values above 0.75, although the mean value is still close to 0.5—. Instead, these basins exhibit a partitioning pattern

closer to $P$-halved. This leads to the question of why the excess water availability in the more forested energy-limited basins does not result in an $R$-dominated pattern. This may be related to the role of forests in regulating the surface water balance, as will be discussed in the next section.



**Figure 4.** Distribution of spatially averaged $E_p$ (a), $E_p/P$ (b) and $k$ (c; same as Fig. 2c) for the 22 basins organized by increasing forest cover fraction (green shade), for the 2001–2012 period. Boxplots describe the spatial variability of $E_p$ (a), $E_p/P$ (b) and $k$ within each basin. Blue lines are the LOESS fittings and grey shades are the corresponding 95% confidence interval.

In summary, the observed distinction between $E$-dominated and $P$-halved partitioning patterns is not equivalent to the distinction between water- and energy-limited environments. Under the perspective of the Budyko hypothesis, *for a given $P$*, an increase of $E_p$ would force the partitioning towards an $E$-dominated pattern, while decreasing $E_p$ should favor the occurrence of an $R$-dominated pattern. The reasons for the occurrence of a $P$-halved pattern are less evident from this perspective, because such a partitioning pattern requires an approximate balance between $E$- and $R$-production processes. These processes,





synthesized by $k$, depend on biophysical mechanisms and basin attributes that are not fully incorporated in the $E_p/P$ ratio. The long-term water balance partitioning depends not only on the available water ($P$) and energy ($E_p$), but also on biophysical processes that are determinant for real evaporation ($E$) and runoff ($R$). Of note is also that $P$ (water availability) is not a given amount of water that is independent of the presence of forests in large basins. This is where the role of forests can become
crucial.

## 4.2 The role of forests

The observed partitioning patterns indicate that $k$ increases with forest cover (in the less forested basins), but then it approximately stabilizes around $k \sim 0.5$ (in the more forested basins; Fig. 2). This leads to the question of whether and how this partitioning patterns are related to the presence of forests. We propose the hypothesis that the presence of forests in large basins
is determinant for the occurrence of a given partitioning pattern, mainly because forests have a strong potential to modify the components of the surface water balance. There is a variety of mechanisms through which forests can exert strong effects on the components of long-term water balance in river basins These factors include, but are not limited to: accumulation and redistribution of soil moisture by root systems (Nadezhdina et al., 2010; Nepstad et al., 1994; Lee et al., 2005; Bond et al., 2002), strong capacity for stomatal regulation related to the large cumulative surface area of leaves (Berry et al., 2010; Costa
and Foley, 1997; Katul et al., 2012), physiological adaptations for water and light use efficiency (Nadezhdina et al., 2010), landscape-scale energy balance effects and overall dynamics of $E$ (Villegas et al., 2014), land-atmosphere interactions that enhance the capacity of river basins to store water as a natural "reservoir" (Salazar et al., 2017), activation of shallow convection through transpiration (Wright et al., 2017), below-canopy stability that restricts direct soil evaporation (Henao et al., Submitted), and soil moisture control via canopy effects on hydrological partitioning (Fleischbein et al., 2005). Collectively,
these mechanisms define biophysical relations between the presence of forests and the dynamics of $P$, $E$ and $R$, and imply a strong potential of forests for modifying the long-term patterns of water balance partitioning.

The long-term effect of forests on the production of $E$ or $R$ is not in a single direction, which is likely related to several features of the observed partitioning patterns. The increase of forest cover in a basin does not always translate into increased $E$ and reduced $R$; the effect may be in the opposite direction as well. This is a consequence of the dual capacity of forests to
either increase or decrease the components of the long-term water balance. For instance, forests can increase or decrease $E$ via opening or closing stomata, respectively. If the effect of forests was always to increase $E$, then the increase of forest cover should be associated with an increase of the relative dominance of $E$ over $R$ (i.e. a reduction of $k$), consistent with a transition from an $R$-dominated to an $E$-dominated pattern. This is contradicted by the observation that in the less forested basins $k$ increases with forest cover (Fig. 2c). Such an increase of $k$ with increasing forest cover may result from the interplay between
a relative reduction of $E$ (via e.g. stomatal regulation, below canopy stability, and aerodynamic resistance associated with the presence of trees), and an associated increase of $R$ linked also with an enhancement of land water storage and base flow (via infiltration).

The increase of $k$ with increasing forest cover is not unlimited: there is not a transition between an $E$-dominated pattern in the less forested basins to an $R$-dominated pattern in the more forested basins. Instead, an approximate balance between the



production of $E$ and $R$ (the $P$-halved pattern) is approached as the forest cover fraction increases (Fig. 2c). The dual capacity of forests to increase or decrease the water balance components implies that the increase of forest cover enhances the capacity of a basin to produce $E$ or $R$. Since $E$ and $R$ are competing water fluxes, such a dual capacity allows for the occurrence of the balance between $E$- and $R$-production processes that is required for the $P$-halved pattern to occur.

5  The approximate balance between $E$ and $R$ in the more forested basins is suggestive of regulation mechanisms acting on the long-term water balance partitioning. The capacity of a river basin for regulating the components of the surface water balance is summarized by its capacity for storing water and controlling its release —this is analogous to the capacity of artificial reservoirs to regulate river flows, which depends on its capacity for storing water and operation rules about how to release it (Magilligan and Nislow, 2005)—. River basins have natural mechanisms to implement these processes of water handling, which depend

10 importantly (but not exclusively) on their geological and geomorphological properties (Bruijnzeel, 2004; Miguez-Macho and Fan, 2012). However, the observation that the $P$-halved pattern is common to basins that differ widely in their geological and geomorphological properties suggests that the occurrence of this pattern is related to other properties. A common feature of basins exhibiting the $P$-halved pattern is that they are mostly covered by forests (forest cover fraction is larger than $\sim 0.5$). The abundance of forests is likely to enhance the natural capacity of large river basins to store water and control its release through

15 land-atmosphere interactions, thereby enhancing the capacity for regulating the components of the water balance —we have proposed the "forest reservoir concept" to described this forest-related regulation capacity (Salazar et al., 2017)—.

  The long-term effect of forests is not only on $E$ and $R$ but also on $P$. Continental precipitation (and therefore water availability in the Budyko framework) is not independent of the presence of forests —among the studied basins, correlation between $P$ and forest cover fraction is 0.61 ($p = 0.003$)—. Different perspectives could be used to explain this relation. One view is that

20 forests tend to grow in regions with relatively high water availability, consistent with observation that the more forested basins are not limited by water but by energy (Fig. 4b). However, this view implicitly assumes that water availability in a river basin (especially the precipitation pattern) precedes (it is the cause for) the existence of forests (the effect) and, therefore, that precipitation is largely independent of the presence of forests itself. This is contradicted by increasing scientific evidence that forest cover change can significantly alter precipitation regimes in many regions of the world (e.g. Mahmood et al., 2014; Spracklen

25 and Garcia-Carreras, 2015; Lawrence and Vandecar, 2015; Zemp et al., 2017), and that land evaporation is a large source for continental precipitation (Van der Ent et al., 2010; Gimeno et al., 2012) whereby forests are major contributors (Bonan, 2008; Schlesinger and Jasechko, 2014). If precipitation regimes were independent of forest-related ecohydrological processes, those regimes should not significantly change in response to forest cover change. We adopt the premise that water availability and forests are directly linked (Ellison et al., 2012), mainly associated with land-atmosphere interactions that define a double-way

30 relation between precipitation regimes and forest-related ecohydrological processes (e.g. evaporation-mediated precipitation recycling). Further, this double-way relationship is particularly important in large basins.

  As a consequence of the potential feedbacks between $P$ and forest-related processes, increased $E$ over forests does not necessarily lead to a long-term reduction in $R$, but rather it can be a component of a transport mechanism that redistributes moisture across the basin. Increased $E$ can enhance upstream (downwind) $P$ through atmospheric moisture transport related to

35 precipitation recycling (Zemp et al., 2017; Makarieva et al., 2013; Spracklen et al., 2012). For instance, a number of modeling





studies consistently indicate that large-scale forest may lead to a reduction of $P$ over the Amazon basin (Spracklen and Garcia-Carreras, 2015), which could result in reductions of $R$ (Stickler et al., 2013; Coe et al., 2009). The potential effects of forest loss on precipitation regimes over the Amazon include a lengthening of the dry season (Costa and Pires, 2010) and disruption of the wet season onset (Wright et al., 2017). All these effects are likely related to a weakening of vegetation-atmosphere

feedbacks due to forest loss (Zemp et al., 2017).

The capacity of forests to increase or decrease the water balance components is also consistent with observation in which the $R$-dominated partitioning pattern is generally absent in more forested basins. Not finding the $R$-dominated pattern indicates that $E$-production is generally dominant across the basins —a usual feature of natural ecosystems; (Huxman et al., 2005)—, with the less dominance when the pattern is $P$-halved. In less forested basins, most of $P$ is converted into $E$ leading to values

of $k$ that approach zero as forest cover fraction reduces, corresponding to water-limited environments. (Shen and Chen, 2010). A reduction of forest cover reduces the natural capacity of a basin to retain water in the surface (including the ecologically-active root zone in the soil), thereby favoring the conversion of available energy ($E_p$) into latent heat ($E$), resulting in a relative reduction of the fraction of $P$ that is potentially converted into $R$. $R$-production (we are considering river runoff after accumulation along the river network, $R = Q/A$) is a slower process that requires the accumulation of runoff through surface

and subsurface flows. In large basins, a characteristic time-scale for $R$-production ranges from $10^{-1}$ to $10^2$ days (or even longer), as given by either the concentration time (e.g. Fang et al., 2008) or the water residence time (e.g. McGuire et al., 2005). As compared to $E$, enhancing $R$ requires a longer time of residence of water in the surface. Forests have a strong potential to enhance this residence time by restraining $E$, as well as by favoring the retention of water within the coupled land-atmosphere system of the basin and its slow routing to river networks (Salazar et al., 2017).

## 5   Concluding remarks

In synthesis, our results highlight the potential occurrence of two dominant patterns ($E$-dominated and $P$-halved) in the long-term water balance partitioning (described by $k$) occurring in large basins of the world. The occurrence of these two patterns largely coincides with the distinction between less forested and more forested basins. The distinction between the $E$-dominated and $P$-halved patterns is related but not fully explained by differences between water- and energy-limitations. Instead, the

occurrence of any specific partitioning pattern in a given basin depends on the biophysical processes and basin attributes that affect $P$, as well as its conversion into either $E$ or $R$. Further, our results indicate that forest cover is an effective descriptor of those basin attributes that are relevant for characterizing long-term water balance partitioning in large basins of the world.

Predicting the environmental response of large basins to forest loss remains a fundamental problems in hydrological sciences today (Ellison et al., 2012; Montanari et al., 2013; Zhou et al., 2015; Zhang et al., 2016). Overall, our results support the

view that the presence of forests enhances continental water availability through an improved capacity in major river basins to maintain large river flows in the long term. A critical implication of our results is that forest loss —related to current unprecedented rates of forest loss due to land use conversion, drought induced tree die-off, deforestation and forest fires (Allen et al., 2015; Davidson et al., 2012)— can force a basin from the $P$-halved to the $E$-dominated partitioning pattern, affecting




the long-term average production of $R$ and, in consequence, river flow regimes that are determinant for many ecological and societal processes (Piao et al., 2007; Coe et al., 2009; Sterling et al., 2013; Lima et al., 2014; Zhang et al., 2016).

## 6 Data availability

Data used for this study are available from the lead author (daniel.mercado@udea.edu.co).

5 *Author contributions.* D. Mercado-Bettín, JC Villegas and JF Salazar designed the research and wrote the manuscript; D. Mercado-Bettín performed data analysis.

*Competing interests.* The authors declare that they have no conflict of interest.

*Acknowledgements.* Funding was provided by "Programa de investigación en la gestión de riesgo asociado con cambio climático y ambiental en cuencas hidrográficas", Convocatoria 543-2011 Colciencias. JCV was partially supported by NSF- EF-1340624 through the University 10 of Arizona. River flow data was partially obtained from IDEAM; Global Runoff Data Centre; and Olga Semenova - Gidrotehproekt Ltd., St. Petersburg State University.





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
