# Peer review of "Global synthesis of forest cover effects on long-term water balance partitioning in large basins"

_Hydrology and Earth System Sciences, 2017_

## Referee Comment (RC1) · Anonymous Referee #1 · 1 Nov 2017

**General comments**

The manuscript "Global synthesis of forest cover effects on long-term water balance partitioning in large basins" used observed data world basins worldwide and found a striking pattern in water balance partitioning depending on forest coverage. The paper is in general well-written and addresses a topic of great interest for the HESS readership. The research question and results are intriguing and elegant. However, there are several issues related to the authors' method explanations and results interpretations that makes the paper an unconvincing read:

- **Unclear explanation for observed relationship between forest cover and**

[Figure]

**river flows.** The authors suggest that forests increases river flows, although their data and analyses do not support this, being neither based on time series in basins undergoing forest cover change nor on analyses of absolute river flow amounts. Such claims are made e.g. at P8L10-11: "our results are consistent with the contrasting view that the presence of forests enhances the long term capacity of river basins to maintain large river flows", and at P13L29-31: "our results support the view that the presence of forests enhances continental water availability through an improved capacity in major river basins to maintain large river flows in the long term". The authors' thought process is not clearly explained. One could imagine that the authors think that an increase in forest cover attracts P (e.g., through the still debated biotic pump mechanism), and therefore a stable P-half pattern would mean increased runoff. However, the authors' explanations in other parts have rather argued that forests can both increase and decrease both E and alter P through moisture recycling. If moisture recycling is the main mechanism, it seems more intuitive to think that forest cover does not have a large influence on river flows at all, since simultaneous changes in E and P in the same direction should result in a dampened change in river flow. The authors makes a long list of different mechanisms through which forest regulate the surface water balance on p. 11, but it is not clearly explained how these processes act together to result in the observed pattern and fall in with the authors' claim that forest cover increase can increase river flows. I would suggest the authors to better guide the reader through their thought process step by step, and include clarifying conceptual diagrams of the processes and interactions addressed.

- **Invalid to substitute space for time.** The authors use their observations in space (i.e. comparison across different river basins) to draw conclusions about how river flow changes with changes in forest cover (i.e., temporal changes in river basin). I would suggest the authors to refrain from making such jumps in their conclusions. If the authors insist to discuss the possibility that spatial

comparisons can be indicative of temporal changes, this limitation needs to be highlighted much more and the level of certainty in the claims need to be toned down. Criticism of space-for-time type of research can for example be found in (Berghuijs and Woods, 2016; Ratajczak and Nippert, 2012). The authors seems to recognize this (e.g., from P8L33), but still jumps into rather spectacular conclusions with formulations like "A critical implications of our results is that forest loss ... []... can force a basin from the P-halved to the E-dominated partitioning pattern" (P13L31).

- **Basin selection rationale unexplained.** The results are highly dependent on the basin selection. Therefore, it is of great importance how basins are selected. With 22 basins, even a relatively small bias in basin selection could seriously affect the results. Please provide information on how the basins were selected.

- **Additional analyses could support a more satisfying explanation.** The manuscript shows a pattern, but does not provide a satisfying explanation. The authors make an attempt to explain the observed pattern through Budyko curve analyses (which did not provide an explanation), and subsequently make several rather speculative explanations in the discussions, pertaining to e.g., moisture recycling (P12L34-35), forest reservoir concept (P12L16), basin size (P12L31) etc. I found it somewhat disturbing that such large parts of the manuscript discussions are solely based on speculative interpretation, rather than discussion of performed analyses results. I think the manuscript would feel more complete, if the authors could perform a few more analyses to test some of these suggested ideas: e.g., what are the approximate moisture recycling ratios in the P-halved basins? How is the P-halved pattern optimal for regulating water flows? Is basin size correlated with the forest's regulation capacity? While true that there are times when interesting observations should be published even when no satisfactory explanation can be put forward, I think there is room for a few more, not overly demanding analyses.

- **Please refer to criticism and controversies.** The paper cites several papers, whose validity is questioned. For example, (Ellison et al., 2012) is mentioned several time throughout the paper and cited unchallenged despite serious issues with the paper have been pointed out by (van der Ent et al., 2012). Another case is (Zhou et al., 2015) that has been criticised by (Berghuijs and Woods, 2016). The biotic pump theory put forward by (Makarieva et al., 2012) is controversial in terms of its very physics (Meesters et al., 2009). Please check.

**Specific comments**

- P1L20. Please provide reference supporting the view that forest can lead to an increase in river flows due to e.g. precipitation recycling.

- P1L21. Please note that (Ellison et al., 2012) is a review paper, and the interpretation of observation/modelling results within have been challenged by (van der Ent et al., 2012).

- P1L16-21. (Wang-Erlandsson et al., 2017) provides a process based explanation of land-use change effects on river flows that includes the moisture recycling mechanism and could be useful to cite.

- P3L26. Please clarify how potential evaporation here is defined and calculated.

- Figure 1 shows some interesting patterns: there are several basins where high forest cover coincide with low R/P ratio (e.g., Lena, Mackenzie); there are basins where runoff ratio seems unaffected by forest cover (e.g., Tapajos); and in e.g., Parana, a high forest cover close to the basin outlet seems to have brought down the k value away from the P halved pattern. Please discuss how this relate to the overall basin wide k value patterns, and how it fits into the narrative of forest cover being the explanatory factor of the P halved pattern.

- The two final "Results" paragraphs starting at P8L24 reads like "Discussions". Please consider re-allocation.

- Section 4.1 reads partly as "Results" rather than "Discussions". Please consider to re-organise.

- At P11L23-24, the authors write "the increase of forest cover in a basin does not always". It is not clear if the authors refer to a temporal change in forest cover (which then should be supported by references) or a comparison between basins in the paper (which then should be supported by a cross-reference and formulated as a comparison rather than an actual "increase").

- P11L23-24. (Teuling et al., 2010) sheds some lights on contrasting hydrological behaviour between European grassland and forests and could be worth citing.

- The authors contradict two views, on p.12 from l. 17. One view is that "forests tend to grow in regions with relatively high water availability", but is "contradicted by the increasing scientific evidence that forest cover change can significantly alter precipitation regimes in many regions of the world". This statement is problematic because the references listed, while showing that forest has the ability to alter precipitation, never contradict the notion that forests tend to grow in regions with high water availability.

**Technical corrections**

- There are in several cases an erroneous em-dash at the end of sentences (e.g., P12L9, P13L8), please check.

- In Fig. 1, the subplots are not always well-aligned. Please check.

**References**

Berghuijs, W. R. and Woods, R. A.: Correspondence: Space-time asymmetry undermines water yield assessment, Nat. Commun., 7, 11603, doi:10.1038/ncomms11603, 2016.

Ellison, D., Futter, M. N. and Bishop, K.: On the forest cover–water yield debate: from demand- to supply-side thinking, Glob. Chang. Biol., 18(3), 806–820, doi:10.1111/j.1365-2486.2011.02589.x, 2012.

van der Ent, R. J., Coenders-Gerrits, M., Nikoli, R., Savenije, H. H. G. and Coenders-Gerrits, A. M. J.: The importance of proper hydrology in the forest cover- water yield debate: commentary on Ellison et al. (2012) Global Change Biology, 18, 806-820, Glob. Chang. Biol., 1–12, doi:10.1111/j.1365-2486.2012.02703.x, 2012.

Makarieva, A. M., Gorshkov, V. G. and Li, B.-L.: Revisiting forest impact on atmospheric water vapor transport and precipitation, Theor. Appl. Climatol., doi:10.1007/s00704-012-0643-9, 2012.

Meesters, A. G. C. A., Dolman, A. J. and Bruijnzeel, L. A.: Comment on "Biotic pump of atmospheric moisture as driver of the hydrological cycle on land" by A. M. Makarieva and V. G. Gorshkov, Hydrol. Earth Syst. Sci., 11, 1013–1033, 2007, Hydrol. Earth Syst. Sci. Discuss., 6(1), 401–416, doi:10.5194/hessd-6-401-2009, 2009.

Ratajczak, Z. and Nippert, J. B.: Comment on "Global resilience of tropical forest and savanna to critical transitions"., Science, 336(6081), 541; author reply 541, doi:10.1126/science.1219346, 2012.

Teuling, A. J., Seneviratne, S. I., Stöckli, R., Reichstein, M., Moors, E., Ciais, P., Luyssaert, S., van den Hurk, B. J. J. M., Ammann, C., Bernhofer, C., Dellwik, E., Gianelle, D., Gielen, B., Grünwald, T., Klumpp, K., Montagnani, L., Moureaux, C., Sottocornola, M. and Wohlfahrt, G.: Contrasting response of European forest and grassland energy exchange to heatwaves, Nat. Geosci., 3(10), 722–727, doi:10.1038/ngeo950, 2010.

Wang-Erlandsson, L., Fetzer, I., Keys, P. W., van der Ent, R. J., Savenije, H. H. G.

and Gordon, L. J.: Remote land use impacts on river flows through atmospheric tele-connections, Hydrol. Earth Syst. Sci. Discuss., 1–17, doi:10.5194/hess-2017-494, 2017.

Zhou, G., Wei, X., Chen, X., Zhou, P., Liu, X., Xiao, Y., Sun, G., Scott, D. F., Zhou, S., Han, L. and Su, Y.: Global pattern for the effect of climate and land cover on water yield, Nat. Commun., 6, 5918, doi:10.1038/ncomms6918, 2015.
* * *

---

## Short Comment (SC1) · 8 Nov 2017

**Peer-review "Global synthesis of forest cover effects on long-term water balance partitioning on large basins"**

**Daniel Mercado-Bettín, Juan F. Salazer, Juan C. Villegas**

By A.C.E. Neefjes

**Note to the editor and authors**
As part of an introductory course to the Master programme Earth & Environment at Wageningen University, students get the assignment to review a scientific paper. Since several years, students have been reviewing papers that are in open online discussion for HESS, and they have been asked to submit their reports to the discussion in order to help the review process. While these reports are written as official (invited) reviews, they were not requested for by the editor, and we leave it up to the editor and authors to use these reports to their advantage. While several students were asked to review the same paper, this was not done with the aim to provide the authors with much extra work. We hope that these reports will positively contribute to the scientific discussion and to the quality of papers published in HESS. This report/review was supervised by dr. Ryan Teuling.

**Short abstract**
Knowledge on the effects of forest cover on water balance partitioning is becoming more and more relevant with ongoing forest loss. Forest cover is shown to be an effective descriptor for characterizing the water balance partitioning in the 22 largest basins around the world. The long-term water balance is evaporation-dominated with both the runoff coefficient and forest cover below 0.5, while evaporation and runoff are equally divided when the forest cover fraction becomes above 0.5. Understanding the responses of the continental water balance on changing forest cover is necessary in a changing environment, but there was still no consensus on this topic. This paper provides new insights in the determination of water balance partitioning using forest cover as a proxy.

**General overview with recommendation**
The article suits the scope of the journal well, since it seeks to understand the interactions between water, ecosystems and alterations of ecosystems by human influence. The objective of the research is clear and researched in a good manner. Furthermore, the methods are clearly written. The best method for water balance partitioning (runoff coefficient) is chosen. Sen and Altunkaynak (2006) investigated different versions of the runoff coefficient and thereby rainfall-runoff simulation and concluded that the runoff coefficient shows the least relative error compared to other methods. Besides, the choice of using different types of statistical methods increases the credibility of the results. In this manner strengths and weaknesses of all types of methods can be taken into account to come to a solid conclusion. Furthermore, using basins all around the world and the complete range of forest cover fractions increases the credibility and usage of the method worldwide. The figures are clearly presented and can directly be understood. The discussion section elaborates greatly on the processes that could be of importance in the relation between forest cover and water balance partitioning. Besides addressing the role of water- or energy-limited systems, also the role of forests is discussed. Because of the broad discussion, the mechanisms behind the results become very clear, which makes the final conclusion more convincing.

However, I am still not fully convinced by the conclusion that forest cover is an effective descriptor of basin attributes describing the long-term water balance for the high forest cover fraction (>0.5). Since 5 out of 14 basins fall outside the 95% confidence interval and these outliers are not discussed in the paper, it does not seem a very good

method to describe the water balance partitioning in these basins. These outliers should be discussed as well in order to become fully convincing. Besides, the authors state that forest loss will force a system into a evaporation-dominated system. However, this is not well explained and other papers do not agree with this statement. Furthermore, the structure of the paper should be improved, in order to make the readability better. Some major revisions need thus to take place in order to be able to accept the paper, which I will elaborate on in the next paragraphs.

**Major revisions**
The largest concern of this article is that the conclusion is not fully supported by the observations. The conclusion that forest is an effective descriptor of the basin attributes that are relevant for characterizing the long-term water balance in large basins of the world is not fully convincing, since the results (figure 2) show that 5 out of 14 basins (Vitim, Madeira, Mackenzie, Lena and Purus) with a high forest cover all outside the 95% confidence curve, which is 35% of all basins. This is a large fraction of basins deviating from the pattern. Besides, Sava and Solimo Jav. are just on the edge of the interval. These basins are not addressed in the results, but only 5 out of 22 basins are discussed that are consistent with the pattern they want to show. Although the broad pattern of the water partitioning and the forest cover has been discussed, the basins that not showing this pattern should be addressed as well.

A possible reason of the deviation could be found by Ellison et al. (2012), who evaluated the existing controversy about the influence of forest cover on the downstream water availability. They concluded through a review that the scale at which the influence of forest cover have been investigated matter. Forest cover influences the global precipitation by the evapotranspiration they produce. Removing forest has transboundary implications for local and global ET and the water regime. Demand-thinking looks only locally into the needs of a forest (water consumption) rather than larger scale at the creation of water supply to another area. Both processes are important and cannot be seen separately. Water supply through evapotranspiration raises precipitation in other regions as well, affecting the water balance partitioning in another catchment. This article put emphasis on the effect of forest cover loss on transboundary regions, which has not been investigated in this paper and could influence the investigated water balance partitioning in the basins.

The lack of explanation of the basins deviating from the general pattern in figure 2 has a big impact on the credibility of the conclusion and should thus be explained well. From the paper it should become clear why these basins are showing different patterns regarding the water partitioning. If there is no clear reason, the conclusion for basins with a high forest fraction is not convincing, since 35% of the basins is deviating from the pattern.

Besides, it is also concluded that the results provide insight in understanding and predicting potential consequences of forest cover loss on the continental water balance. It is addressed that forest loss can force a basin to go from P-halved to E-dominated, affecting the production of runoff and thus the river flow regimes. This conclusion is not fully supported as well. The first contradiction can be found in the discussion (line 23, page 11), stating that forest cover increase will not lead to increased evaporation and decreased runoff, but can be the other way around as well. This is in agreement with line 33, page 8 in the observations, that states that the relations between the partitioning patterns and forest cover are intended to be only descriptive and not predictive. Furthermore, Zhang et al. (2016) investigated the hydrological responses of forest cover change and to influence of spatial scale, climate, forest type and hydrological regime in large and small watersheds around the world. They achieved this by investigating the

response intensity of annual runoff to forest cover change. Climate conditions were investigated through the Budyko Dryness Index in order to determine whether a watershed was energy or water-limited. Besides, they compared different forest cover types (broadleaf, mixed and coniferous) and hydrological regimes (rain and snow-dominated) to test for their significance. They concluded that an increase in annual runoff due to forest cover decrease is significant at several spatial scales.

All in all, the discussion and observations and the paper thus concluded the opposite from the article's conclusion that a forest will change into an evaporation-dominated system. More explanation should be given in order to make the article's conclusion more convincing. It should become clear why these results give insight in the consequences of forest loss and why is will go into an evaporation-dominated system rather than runoff-dominated.

The last major concern is that spurious correlations could be present between the runoff coefficient and forest cover due to transboundary effects that are not taken into account in his paper. Although the runoff coefficient is shown to be the best method to partition the water balance (Sen and Altunkayak, 2006), this method does not take transboundary effects into account. Ellison et al. (2012) showed that these effects are relevant in forested basins. Other papers that put emphasis on this are e.g. Ellison et al. (2017) who reviewed a substantial amount of research on forest, water and energy interactions and Debortoli et al. (2016) who investigated the relationship between forest cover and rainfall patterns in the Amazon Forest using rainfall timeseries between 1971 – 2010 and forest cover from LANDSAT5 satellite data. They found that on regional scale forest cover and seasonal rainfall are correlated. Furthermore, forest loss resulted in a decrease in evapotranspiration and important implications for rainfall thousands of kilometres downwind. Focussing on the runoff coefficient only evaluates the effect of either changes in precipitation, runoff or evaporation. It is possible due to transboundary effects that the results can be affected by more precipitation in the upstream parts of the catchment that also has a large forest cover (Ellison et al., 2017). Forest could thus influence the water balance partitioning of another basin as well, resulting in more complex relationships between forest loss and the runoff coefficient. Spurious correlation between forest cover and runoff should be excluded. This can be done by showing that the same results can be obtained using average evapotranspiration (precipitation minus runoff). Otherwise, the signal of the correlation found has not been identified clearly.

Finally, some minor issues should be addressed.

**Minor Revisions**

- The title should be more attractive. Now it is too general and does not include the main conclusion of the article. This could be changed into: "Forest cover as descriptor for long-term water balance partitioning".
- The structure of the article should be revised. In the discussion part 4.1 figure 4 is introduced, while in the methods they elaborate on the way how they obtained this data. In my opinion, this should be included in the observations and compared there, while the reason behind the possible connections should be discussed in the discussion. In the discussion session the observations should be discussed, while in the observation session no discussion should be started. Both parts of the paper should be clearly distinctive. Introducing new observations in the discussion is confusing, because one expects to already have received all information on the research done in this study. It is better to transfer the observations (line 10-18 and 26-30 of page 9) to the observational part of the paper and to transfer the challenges and discussion of the patterns showed in the observation part (line 24-

35 of page 8 and line 1-7 of page 9) to the discussion part, because this can be seen as recommendations for further research. However, what also could be done in order to be more clear, is to change the title of section 3 to something like: "Patterns in water balance partitioning" and section 4.1 should become 3.2, discussing the effect of water- and energy limitations. Then the discussion should only consist of the role of forests, bringing all observations together. However, the recommendations (line 24-35, page 8 and line 1-7, page 9) should always be discussed in the discussion section rather than the results. It is up to the authors to decide which method to use.

- Argument why a particular method is used. More emphasis should be given to the reason why the runoff coefficient rather than another method is chosen. Furthermore, it should explained why they used a 12-year average for the calculation of the k-value and not another timeframe. It is stated that forest cover is decreasing, it would also be interesting to investigate the changes in partitioning throughout the years and see what influence this change has on the water balance. Besides, a metric of forest is provided by constructing a global land cover map using the temporal mode for each pixel in 2001-2012 map series of MODIS-MCD12C1 with five land use classes. They constructed the land use maps in temporal mode, because landcover is not static but changing through time. This seems logic. However, since they used the runoff coefficient averaged of 12 years, it is not clear why to use temporal landcover data. Furthermore, they divided the land uses into five classes using the 16 classes of IGBP and only investigated forest or non-forested areas. They could also have chosen to investigate the influence of other land use types as well dividing this way, since they have a significant impact on the runoff generation (Mahe et al., 2004). With the division, they put all forested areas in one class, so including evergreen and deciduous. Explanation of the choices made will clarify these questions.
- Ellison et al. (2017) should be included, because this paper reviews on the latest research done on the effect of forest cover on the water balance and vice versa.
- Page 1, line 21: "For instance" can be deleted.
- Page 2, line 1: change "for" to "in".
- Page 2, line 1-2: "Panta Rhei – Everything flows" debate is not known by everyone and there should be elaborated on this.
- Page 2, line 10-11, In the Amazon ~40% of precipitation is recycled. In the paper they cited an article from 1994 (Eltahir and Bras, 1994). This is relatively old and the number could have changed in the meantime, Bosilivich and Chern (2006) state that the recycling ratio is 27.2%. It could thus have changed between 1994 and 2006, decreasing the recycling ratio by 13%. This should be changed.
- Page 2, line 22: change "otherwise" in "the other way around".
- Page 2, line 30: change "We choose to focus on" in "We focused on"
- Page 3, line 3: delete "," in "precipitation, (P)"
- Page 3, line 20-21: include the units of R, Q and A.
- Page 3, line 27-30: provide the assumptions made using the snow-melt equivalent and the consequences of this.
- Page 7: change "either" in "neither".
- Page 9, line 30: delete "of".
- Page 11, line 8: change "this" in "these".
- Page 11, line 9: delete "the hypothesis".
- Page 13, line 10: delete "." after "environments".

**References**

Bosilovich, M. G., & Chern, J. D. (2006). Simulation of water sources and precipitation recycling for the MacKenzie, Mississippi, and Amazon River basins. *Journal of Hydrometeorology*, *7*(3), 312-329.

Debortoli, N. S., Dubreuil, V., Hirota, M., Lindoso, D. P., & Nabucet, J. (2017). Detecting deforestation impacts in Southern Amazonia rainfall using rain gauges. *International Journal of Climatology*, *37*(6), 2889-2900.

Ellison, D., Morris, C. E., Locatelli, B., Sheil, D., Cohen, J., Murdiyarso, D., ... & Gaveau, D. (2017). Trees, forests and water: Cool insights for a hot world. *Global Environmental Change*, *43*, 51-61.

Ellison, D., N Futter, M., and Bishop, K.: On the forest cover–water yield debate: from demand-to supply-side thinking, Global Change Biology, 18, 806–820, 2012.

Eltahir, E. A., & Bras, R. L. (1994). Precipitation recycling in the Amazon basin. *Quarterly Journal of the Royal Meteorological Society*, *120*(518), 861-880.

Mahe, G., Paturel, J. E., Servat, E., Conway, D., & Dezetter, A. (2005). The impact of land use change on soil water holding capacity and river flow modelling in the Nakambe River, Burkina-Faso. *Journal of Hydrology*, *300*(1), 33-43.

Şen, Z., & Altunkaynak, A. (2006). A comparative fuzzy logic approach to runoff coefficient and runoff estimation. Hydrological Processes, 20(9), 1993-2009.

Zhang, M., Liu, N., Harper, R., Li, Q., Liu, K., Wei, X., Ning, D., Hou, Y., and Liu, S.: A global review on hydrological responses to forest change across multiple spatial scales: importance of scale, climate, forest type and hydrological regime, Journal of Hydrology, 2016.

---

## Referee Comment (RC2) · Anonymous Referee #2 · 17 Nov 2017

This manuscript presents a very interesting hypothesis about the impact of forest cover on long-term partitioning of water between evapotranspiration and runoff for 22 large basins around the world. However, as I outline below I believe the analysis, data and methods require further explanation and revision to justify publication and to strengthen the case for the proposed hypothesis. I have chosen not to comment on the speculated causes of the proposed hypothesis as I believe this would best be done once the observational basis of the hypothesis is stronger.

Major comments

Analysis: The key figure in this manuscript is Figure 2c (repeated in 4c), which presents

a gradual increase in runoff ratio with increasing average forest cover until the runoff ratio reaches ∼0.5. This figure is the basis of the proposed hypothesis. In the figure the individual catchment runoff ratios are presented as box-plots by basin and the basins are ordered by increasing average fraction of forest cover. It is not clear which two variables the LOESS smooth is applied to – one variable is the runoff ratio values, but the other variable could be either average forest cover for each basin or a dummy variable to indicate the different basins. My concern with Figure 2c is that the apparent levelling off of runoff ratio to ∼0.5 when the fraction of forest cover reaches ∼0.5 may be an artefact of grouping the runoff ratios by basin. I think a more convincing presentation of this data would be to plot each catchment individually, rather than group the catchments by basin, as each basin contains catchments that have a range of runoff ratios, forest covers, aridity (potential ET / P), P and R. Plotting each catchment on XY scatterplots of runoff ratio vs forest fraction and runoff ratio vs aridity (coloured by forest fraction) would remove the possibility of an artificial grouping influencing the results. I also think the plot of runoff ratio vs aridity (coloured by forest fraction) could present strong evidence to support, or contradict, the proposed hypothesis that high forest cover results in an even split of P between E and R. In this plot, if for a given aridity the runoff ratio is observed to increase with increasing forest cover then this would support the current conclusions of this manuscript. However, if for a given aridity the runoff ratio is observed to be unrelated to forest cover then this would not support the current conclusions of this manuscript. I think it is very important to compare runoff ratio and forest cover for catchments with similar aridity, to remove confounding the comparison by mixing water and energy limited catchments together.

Data: I have several concerns about the data used in this study outlined below.

River regulation: The results presented in the main body of the manuscript are based on catchments that include regulated and heavily modified catchments. The authors do provide a set of largely similar results for "free flowing" catchments in the Supplementary Material. However, given the aim of the manuscript is to understand the role

of forest cover on long-term partitioning of water in catchments, I would have expected that only "free flowing" catchments would be used in this analysis. The use of regulated or heavily modified catchments adds an extra level of uncertainty to the results that is best avoided. Since the authors have free flowing catchments, I strongly recommend they base their analysis on those rivers only.

Precipitation data: The authors use TRMM-3B42 and ERA-Interim reanalysis data to estimate mean annual precipitation for the period 2001-2012. However, the authors do not cite any evidence that these data sets are consistent with catchment average precipitation estimates based on observed station data for the catchments investigated. How representative are these two products of catchment average precipitation for these catchments?

Snow-melt equivalent: discharge data were modified for snow-melt equivalent in three basins (Mackenzie, Lena & Vitim). How was the snow-melt equivalent discharge identified? The contribution of snow-melt to mean annual runoff in these catchments could be very high. Even if the contribution of snow-melt equivalent can be estimated accurately, I am not convinced that removing the influence of snow-melt from these catchments is reasonable for this analysis. The presence, or absence, of forest cover influences snow accumulation and melting, so forest cover plays a role in the long-term water balance of catchments that receive snow. The role of forest cover in catchments that receive snow should not be ignored in a global synthesis, so I recommend that the influence of snow-melt equivalent not be removed from the discharge data. Accepting this recommendation would also remove the issue of how to identify snow-melt equivalent discharge.

Catchment area: the discharge data from the various source data sets will have a reported catchment area for each catchment. However, the precipitation and potential evaporation data are estimated for catchment areas derived from GTOPO30 and STRM DEMs. Deriving catchment areas from these products is perfectly reasonable. However, it is important to report whether the DEM based catchment area differs from

the reported catchment area associated with the discharge data. As the discrepancy between the DEM and reported areas increases, the precipitation and potential evaporation data becomes less representative of the true area over which the discharge was generated. How large is this discrepancy? If it is <5% then that would be re-assuring. If it is >10% then that would call into question whether the data from that catchment should be used in the analysis.

Selection of basins: I believe the hypothesis should be tested over a wider selection of catchments, particularly catchments in energy limited environments. If largely forested catchments in energy limited environments demonstrate runoff ratios ~0.5 then the evidence for the hypothesis would be more convincing.

Minor comments

Page 3: please note that the potential evaporation estimate from GLEAM v3.0a is based on Priestly-Taylor.

Figure 1: the Sava river has a runoff ratio (k) approaching 1 – is this physically realistic? I suspect not.

Page 7, line 12: "receive a P-input that exhibits small variability and a similar mean value" – it is important to clarify that this statement relates to the small variability in mean annual precipitation between the catchments within the basin. This region of Australia actually receives precipitation with high interannual variability, so it is important to be clear about which variability is being discussed.

---

## Author Comment (AC1) · 18 Dec 2017

Response to Anonymous Referee 1
"The manuscript "Global synthesis of forest cover effects on long-term water balance partitioning in large basins" used observed data world basins worldwide and found a striking pattern in water balance partitioning depending on forest coverage. The paper is in general well-written and addresses a topic of great interest for the HESS reader-

ship. The research question and results are intriguing and elegant. However, there are several issues related to the authors' method explanations and results interpretations that makes the paper an unconvincing read:"

*Thank you very much for your detailed and challenging review. All the suggestions and corrections were accepted. We also clarify some relevant concerns.*

**General comments**

**"Unclear explanation for observed relationship between forest cover and river flows."** The authors suggest that forests increases river flows, although their data and analyses do not support this, being neither based on time series in basins undergoing forest cover change nor on analyses of absolute river flow amounts. Such claims are made e.g. at P8L10-11: "our results are consistent with the contrasting view that the presence of forests enhances the long term capacity of river basins to maintain large river flows", and at P13L29-31: "our results support the view that the presence of forests enhances continental water availability through an improved capacity in major river basins to maintain large river flows in the long term.""

*We agree that our results are not directly supportive of the conclusion that forests increase or enhance river flows. Rather, our main purpose is to show a general pattern in which forest cover can be a descriptor of water balance partitioning. The mentioned sentence at P8L10-11 was written to highlight that our results, in general, do not support the "demand-side thinking" explained in Ellison et al. (2012). However, we agree that this sentence must be changed to avoid misunderstandings. A similar change must be applied to the mentioned sentence at P13L29-31. In the revised version of the manuscript, we modify both sentences to highlight that "the presence of forests coincides with a P-halved water balance partitioning (i.e. the amount of precipitation converted into streamflow is higher in more forested basins).*

"The authors' thought process is not clearly explained. One could imagine that the authors think that an increase in forest cover attracts P (e.g., through the still debated

biotic pump mechanism), and therefore a stable P-half pattern would mean increased runoff. However, the authors' explanations in other parts have rather argued that forests can both increase and decrease both E and alter P through moisture recycling. If moisture recycling is the main mechanism, it seems more intuitive to think that forest cover does not have a large influence on river flows at all, since simultaneous changes in E and P in the same direction should result in a dampened change in river flow. The authors makes a long list of different mechanisms through which forest regulate the surface water balance on p. 11, but it is not clearly explained how these processes act together to result in the observed pattern and fall in with the authors' claim that forest cover increase can increase river flows. I would suggest the authors to better guide the reader through their thought process step by step, and include clarifying conceptual diagrams of the processes and interactions addressed."

*We accept the suggestion of improving our thought process through a better guidance for the reader. To properly achieve that, we will add a "clarifying conceptual diagram" (below) to support the possible explanations of our results and discussion. It is necessary to clarify that we are not supporting that precipitation recycling is the only or the main mechanism responsible for producing the water balance-partitioning pattern that we describe. We generally refer in the text to precipitation recycling as an example of a well-known mechanism that could influence this pattern. In the revised version of the manuscript, lines P2L6, P1L20 and P12L35 will be changed to avoid misunderstandings.*

*To explain how these processes (p. 11) "act together to result in the observed pattern" we will add a new conceptual model that highlights how basins located in arid and semiarid regions are usually controlled by water limited conditions, making radiation the dominant energy and therefore, resulting in a E-dominated partitioning; in basins where significant physical factors control drainage, for instance, a combination of abrupt topography and continuously snowmelt processes, the dominant energy is gravitational, probably resulting in a R-dominated partitioning. Finally, in forested*

*basins, the combination of all physical and ecological processes associated with the presence and function of forests (p. 11), favor a continuous combination of both types of energy (radiation and gravitational) that ultimately result in a P-halved pattern, as illustrated by the data (new conceptual figure).*

**"Invalid to substitute space for time."** "The authors use their observations in space (i.e. comparison across different river basins) to draw conclusions about how river flow changes with changes in forest cover (i.e., temporal changes in river basin). I would suggest the authors to refrain from making such jumps in their conclusions. If the authors insist to discuss the possibility that spatial comparisons can be indicative of temporal changes, this limitation needs to be highlighted much more and the level of certainty in the claims need to be toned down. Criticism of space-for-time type of research can for example be found in (Berghuijs and Woods, 2016; Ratajczak and Nippert, 2012). The authors seems to recognize this (e.g., from P8L33), but still jumps into rather spectacular conclusions with formulations like "A critical implications of our results is that forest loss . . . []. . . can force a basin from the P-halved to the E-dominated partitioning pattern" (P13L31)."

*We agree and accept the suggestion to remove/change the sentences that could wrongly imply that time changes are explained from spatial observations. Accordingly, in the revised version, we modify/remove P13L31-35 from the conclusion section to say that "Our results illustrate that, under particular physical and ecological conditions, forest loss can affect water balance partitioning, such that the general P-halved pattern that we document can be altered".*

**"Basin selection rationale unexplained."** "The results are highly dependent on the basin selection. Therefore, it is of great importance how basins are selected. With 22 basins, even a relatively small bias in basin selection could seriously affect the results. Please provide information on how the basins were selected."

*In the revised version of the text, we include further detail on how basins were se-*

*lected. In particular we clarify that "basins were selected based on: (i) basin sizes large enough to account for internal spatial variations, (ii) availability of discharge data and reliable land cover data". Additionally, we used a number of other basins to test the same patterns, and found similar results, which are presented in the appendix section. Further, we add new figures in the revised version of the manuscript to support our results.*

  **"Additional analyses could support a more satisfying explanation."** "The manuscript shows a pattern, but does not provide a satisfying explanation. The authors make an attempt to explain the observed pattern through Budyko curve analyses (which did not provide an explanation), and subsequently make several rather speculative explanations in the discussions, pertaining to e.g., moisture recycling (P12L34-35), forest reservoir concept (P12L16), basin size (P12L31) etc. I found it somewhat disturbing that such large parts of the manuscript discussions are solely based on speculative interpretation, rather than discussion of performed analyses results. I think the manuscript would feel more complete, if the authors could perform a few more analyses to test some of these suggested ideas: e.g., what are the approximate moisture recycling ratios in the P-halved basins? How is the P-halved pattern optimal for regulating water flows? Is basin size correlated with the forest's regulation capacity? While true that there are times when interesting observations should be published even when no satisfactory explanation can be put forward, I think there is room for a few more, not overly demanding analyses"

*We agree that additional analyses to support our results must be added in the revised version of the manuscript. Following both suggestions, yours and Referee 2, we add new figures to support our results and discussion. Along with the new results commented before and in the response to reviewer 2, we also added a new figure (below) showing the moisture recycling ratios calculated for the 22 basins based on Berger et al (2014). This variable is called "Basin internal evaporation recycling (BIER) ratios", which denotes "the fractions of evaporated water returning to the originating basins via*

*precipitation." (Figure 2 in Berger et al (2014)). The other analysis suggested by the reviewer are also considered in the revised manuscript by either integrating them or commenting on them.*

**"Please refer to criticism and controversies."**  "The paper cites several papers, whose validity is questioned. For example, (Ellison et al., 2012) is mentioned several time throughout the paper and cited unchallenged despite serious issues with the paper have been pointed out by (van der Ent et al., 2012). Another case is (Zhou et al., 2015) that has been criticised by (Berghuijs and Woods, 2016). The biotic pump theory put forward by (Makarieva et al., 2012) is controversial in terms of its very physics (Meesters et al., 2009). Please check."

*Thank you for the suggestion. In the revised version of the manuscript, we explicitly comment on the controversies around these papers and use the suggested references to support the criticisms.*

**Specific comments**

"P1L20. Please provide reference supporting the view that forest can lead to an increase in river flows due to e.g. precipitation recycling." *As we clarified before, this sentence is being adjusted/removed.*

"P1L21. Please note that (Ellison et al., 2012) is a review paper, and the interpretation of observation/modelling results within have been challenged by (van der Ent et al., 2012)." *Corrected as commented in the last comment*

"P1L16-21. (Wang-Erlandsson et al., 2017) provides a process based explanation of land-use change effects on river flows that includes the moisture recycling mechanism and could be useful to cite." *Included in the revised version*

"P3L26. Please clarify how potential evaporation here is defined and calculated." *Clarification is added to the revised version of the manuscript*

"Figure 1 shows some interesting patterns: there are several basins where high forest

cover coincide with low R/P ratio (e.g., Lena, Mackenzie); there are basins where runoff ratio seems unaffected by forest cover (e.g., Tapajos); and in e.g., Parana, a high forest cover close to the basin outlet seems to have brought down the k value away from the P halved pattern. Please discuss how this relate to the overall basin wide k value patterns, and how it fits into the narrative of forest cover being the explanatory factor of the P halved pattern." *As commented before, several other features can also affect water balance partitioning, all related to the dominant form of energy explaining water transit through the basin. These is synthesized in a new conceptual figure described above. Such is the case of individual basins that do not follow the expected trend. A note on these particular examples highlighted by the reviewer is included in the revised version the manuscript.*

"The two final "Results" paragraphs starting at P8L24 reads like "Discussions". Please consider re-allocation." *Corrected as suggested*

"Section 4.1 reads partly as "Results" rather than "Discussions". Please consider to re-organise." *Corrected as suggested*

"At P11L23-24, the authors write "the increase of forest cover in a basin does not al-ways". It is not clear if the authors refer to a temporal change in forest cover (which then should be supported by references) or a comparison between basins in the paper (which then should be supported by a cross-reference and formulated as a compari-son rather than an actual "increase")." *Corrected as suggested in the "space for time substitution" comment.*

"P11L23-24. (Teuling et al., 2010) sheds some lights on contrasting hydrological be-haviour between European grassland and forests and could be worth citing." *Reference added as suggested*

"The authors contradict two views, on p.12 from l. 17. One view is that "forests tend to grow in regions with relatively high water availability", but is "contradicted by the increasing scientific evidence that forest cover change can significantly alter precipita-

tion regimes in many regions of the world". This statement is problematic because the references listed, while showing that forest has the ability to alter precipitation, never contradict the notion that forests tend to grow in regions with high water availability" *Corrected. In the revised version, we avoid the use of the term "contradict".*

**Technical corrections**

"There are in several cases an erroneous em-dash at the end of sentences (e.g., P12L9, P13L8), please check." *We went through the entire manuscript to address these issues.*

"In Fig. 1, the subplots are not always well-aligned. Please check." *Corrected*

**Reference**

Van der Ent, R. J., Savenije, H. H., Schaefli, B., Steele‐Dunne, S. C. (2010). Origin and fate of atmospheric moisture over continents. Water Resources Research, 46(9).

Savenije, H. H. (1996). The runoff coefficient as the key to moisture recycling. Journal of Hydrology, 176(1-4), 219-225.

Ellison, D., N Futter, M., Bishop, K. (2012). On the forest cover–water yield debate: from demand‐to supply‐side thinking. Global Change Biology, 18(3), 806-820.

Berger, M., van der Ent, R., Eisner, S., Bach, V., Finkbeiner, M. (2014). Water accounting and vulnerability evaluation (WAVE): considering atmospheric evaporation recycling and the risk of freshwater depletion in water footprinting. Environmental science technology, 48(8), 4521-4528.
* * *
[Figure]

| Basin type | Physical/ecological processes | Dominant energy | Partitioning type |
|---|---|---|---|

Arid region ➡ Water-limited conditions ➡ Radiation ➡ E-dominated

Drainage-controlled ➡ Topography Snowmelt ➡ Gravitational ➡ R-dominated

Forested

LAI Albedo ➡ Radiation

Infiltrability Water retention ➡ Gravitational

Competence

P-halved

**Fig. 1.** Conceptual Model

**Fig. 2.** Moisture recycling ratios

---

## Author Comment (AC2) · 18 Dec 2017

Response to Anonymous Referee 2
"This manuscript presents a very interesting hypothesis about the impact of forest cover on long-term partitioning of water between evapotranspiration and runoff for 22 large basins around the world. However, as I outline below I believe the analysis, data and methods require further explanation and revision to justify publication and to strengthen

the case for the proposed hypothesis. I have chosen not to comment on the speculated causes of the proposed hypothesis as I believe this would best be done once the observational basis of the hypothesis is stronger."

*Thank you for your interest in our manuscript and your suggestions. The additional suggested analyses were accepted and added in the new version of the manuscript. We also clarify some concerns related to data and methods.*

**Major comments**

"Analysis: The key figure in this manuscript is Figure 2c (repeated in 4c), which presents a gradual increase in runoff ratio with increasing average forest cover until the runoff ratio reaches âĹij0.5. This figure is the basis of the proposed hypothesis. In the figure the individual catchment runoff ratios are presented as box-plots by basin and the basins are ordered by increasing average fraction of forest cover. It is not clear which two variables the LOESS smooth is applied to – one variable is the runoff ratio values, but the other variable could be either average forest cover for each basin or a dummy variable to indicate the different basins. My concern with Figure 2c is that the apparent levelling off of runoff ratio to âĹij0.5 when the fraction of forest cover reaches âĹij0.5 may be an artefact of grouping the runoff ratios by basin. I think a more convincing presentation of this data would be to plot each catchment individually, rather than group the catchments by basin, as each basin contains catchments that have a range of runoff ratios, forest covers, aridity (potential ET / P), P and R. Plotting each catchment on XY scatterplots of runoff ratio vs forest fraction and runoff ratio vs aridity (coloured by forest fraction) would remove the possibility of an artificial grouping influencing the results. I also think the plot of runoff ratio vs aridity (coloured by forest fraction) could present strong evidence to support, or contradict, the proposed hypothesis that high forest cover results in an even split of P between E and R. In this plot, if for a given aridity the runoff ratio is observed to increase with increasing forest cover then this would support the current conclusions of this manuscript. However, if for a given aridity the runoff ratio is observed to be unrelated to forest cover then this

would not support the current conclusions of this manuscript. I think it is very important to compare runoff ratio and forest cover for catchments with similar aridity, to remove confounding the comparison by mixing water and energy limited catchments together."

*Thank you for your appropriate and accurate suggestion. The LOESS smooth was applied to the mean values of k for the 22 basins (P4L20), to analyze possible general patterns in the k-values of the basins. We are aware that internal (to each basin) variations among k values are normal (P4L12-15). Our overall intention in the paper was to assess the effects of forest cover on large basins, even when internal variability in water balance partitioning occurs within them. Consequently, we decided to group sub-basins within larger basin categories. Yet, following the reviewer's suggestion, we implemented the same analysis without grouping the data, and similar results were found. According to your suggestion and to support our results and discussion, we added 4 new figures (or one single panel-plot) using individual catchments to the revised version of the manuscript: 2 smooth-plots containing "runoff ratio vs forest fraction and runoff ratio vs aridity"; and 2 smooth-plots between forest cover vs. k in both, energy-limited and water-limited basins. The addition of these new figures improves clarification and explanation to our results, and provides improved support to our proposed hypothesis. We produced the figures using both the initial basins considered in the paper as well as only free flowing river basins. The results are, in general, consistent in both cases.*

*4 important facts of each figure:*

- *The figures runoff/rainfall vs. Forest support the pattern found in Figures 2 and 3 in the manuscript. They are showing the E-dominated and P-halved patterns.*

- *The figures K (runoff/rainfall) vs PHI (ETP/Rainfall) reflect that the Budyko based hypothesis is not accounting to all processes responsible of the pattern found.*

- *The figures runoff/rainfall vs. Forest using only energy-limited basins*

*(ETP/rainfall<1) support the found patterns (E-dominated and P halved).*

- *The figures runoff/rainfall vs. Forest using only water-limited basins (ETP/rainfall>1), in general, support the found patterns (E-dominated and P halved). The pattern is not so clear when using free-flowing rivers. This is likely related to the fact that there are no basins representing forest fractions between 0.13-0.25 (this is not the case when using the original data).*

"Data: I have several concerns about the data used in this study outlined below. River regulation: The results presented in the main body of the manuscript are based on catchments that include regulated and heavily modified catchments. The authors do provide a set of largely similar results for "free flowing" catchments in the Supplementary Material. However, given the aim of the manuscript is to understand the role of forest cover on long-term partitioning of water in catchments, I would have expected that only "free flowing" catchments would be used in this analysis. The use of regulated or heavily modified catchments adds an extra level of uncertainty to the results that is best avoided. Since the authors have free flowing catchments, I strongly recommend they base their analysis on those rivers only."

*We selected these 22 basins (some of them, heavily modified basins) because the mainstream is big enough to analyze the spatial behavior of k among the basins. However, we understand your concern in the way that, using these data could "add an extra level of uncertainty". Accordingly, following your recommendation, the revised version of the manuscript includes only free-flowing rivers in the main manuscript, while the entire data set is presented in the supplementary material.*

"Precipitation data: The authors use TRMM-3B42 and ERA-Interim reanalysis data to estimate mean annual precipitation for the period 2001-2012. However, the authors do not cite any evidence that these data sets are consistent with catchment average precipitation estimates based on observed station data for the catchments investigated. How representative are these two products of catchment average precipitation for these

catchments?"

*In the revised version of the manuscript, we include appropriate references to support the use of each particular precipitation dataset on each region.*

"Snow-melt equivalent: discharge data were modified for snow-melt equivalent in three basins (Mackenzie, Lena Vitim). How was the snow-melt equivalent discharge identified? The contribution of snow-melt to mean annual runoff in these catchments could be very high. Even if the contribution of snow-melt equivalent can be estimated accurately, I am not convinced that removing the influence of snow-melt from these catchments is reasonable for this analysis. The presence, or absence, of forest cover influences snow accumulation and melting, so forest cover plays a role in the long-term water balance of catchments that receive snow. The role of forest cover in catchments that receive snow should not be ignored in a global synthesis, so I recommend that the influence of snow-melt equivalent not be removed from the discharge data. Accepting this recommendation would also remove the issue of how to identify snow-melt equivalent discharge."

*In the original version, we had decided to subtract the snow-melt equivalent (accumulated in winter) from discharge on these basins to exclude the potential regulatory effect that this process has on base-flow. However, we agree that forest influences snow accumulation and melting, exerting effects on the long-term water balance. Following the reviewer's recommendation, in the revised version, we do not remove the snow-melt equivalent from discharge.*

"Catchment area: the discharge data from the various source data sets will have a reported catchment area for each catchment. However, the precipitation and potential evaporation data are estimated for catchment areas derived from GTOPO30 and STRM DEMs. Deriving catchment areas from these products is perfectly reasonable. However, it is important to report whether the DEM based catchment area differs from the reported catchment area associated with the discharge data. As the discrepancy

between the DEM and reported areas increases, the precipitation and potential evapo-
ration data becomes less representative of the true area over which the discharge was
generated. How large is this discrepancy? If it is $< 5\%$ then that would be re-assuring.
If it is $> 10\%$ then that would call into question whether the data from that catchment
should be used in the analysis."

*Thank you for your suggestion. We made sure that all differences between the source
data and calculated drainage areas were lower than $10\%$. In addition, as a part of the
basin delimitation process, we adjusted the DEMs in some regions with large flat areas
(e.g, the Amazon basin), using the original streamflow network from the data source,
such that errors were minimized. These clarifications are included in the revised ver-
sion of the manuscript.*

"Selection of basins: I believe the hypothesis should be tested over a wider selection of
catchments, particularly catchments in energy limited environments. If largely forested
catchments in energy limited environments demonstrate runoff ratios âĹij0.5 then the
evidence for the hypothesis would be more convincing."

*We agree with the reviewer that more data would be desirable to further improve the
strength of our conclusions. We use the largest collection of basins with topographic,
climatic, hydrologic and vegetation cover data available for us when the analysis were
performed. We include multiple data sources (as described in the methods section). In
the revised version, we include only free-flowing river basins and, therefore, the number
of basins decreases. However, even with a more limited number of basins, the main
observations remain valid, which leads us to believe that when more data becomes
available, the overall behavior should hold.*

**Minor comments**

"Page 3: please note that the potential evaporation estimate from GLEAM v3.0a is
based on Priestly-Taylor." *Clarification added in the revised version*

"Figure 1: the Sava river has a runoff ratio (k) approaching 1 – is this physically realistic? I suspect not." *Thank you for the observation. This number was derived from the original data. However, since it seems physically unrealistic, we excluded this basin from the analysis.*

"Page 7, line 12: "receive a P-input that exhibits small variability and a similar mean value" – it is important to clarify that this statement relates to the small variability in mean annual precipitation between the catchments within the basin. This region of Australia actually receives precipitation with high interannual variability, so it is important to be clear about which variability is being discussed." *We clarify in the revised version that we refer to spatial variability, as highlighted by the reviewer.*

––––––––––––––––––––––––

[Figure]

**Fig. 1.** Panel with original rivers

[Figure]

Energy limited (PHI<1)

Water limited (PHI>1)

**Fig. 2.** Panel with free flowing rivers

---

## Author Comment (AC3) · 18 Dec 2017

Response to A.C.E. Neefjes

Note to the editor and authors

As part of an introductory course to the Master programme Earth Environment at Wageningen University, students get the assignment to review a scientific paper. Since several years, students have been reviewing papers that are in open online discussion for HESS, and they have been asked to submit their reports to the discussion in order to help the review process. While these reports are written as official (invited) reviews,

they were not requested for by the editor, and we leave it up to the editor and authors to use these reports to their advantage. While several students were asked to review the same paper, this was not done with the aim to provide the authors with much extra work. We hope that these reports will positively contribute to the scientific discussion and to the quality of papers published in HESS. This report/review was supervised by dr. Ryan Teuling.

*Thank you very much for your comments, they are certainly helpful for improving its scope, clarity and discussion points. We appreciate your interest in our manuscript*

**Short abstract**

Knowledge on the effects of forest cover on water balance partitioning is becoming more and more relevant with ongoing forest loss. Forest cover is shown to be an effective descriptor for characterizing the water balance partitioning in the 22 largest basins around the world. The long-term water balance is evaporation-dominated with both the runoff coefficient and forest cover below 0.5, while evaporation and runoff are equally divided when the forest cover fraction becomes above 0.5. Understanding the responses of the continental water balance on changing forest cover is necessary in a changing environment, but there was still no consensus on this topic. This paper provides new insights in the determination of water balance partitioning using forest cover as a proxy.

**General overview with recommendation**

The article suits the scope of the journal well, since it seeks to understand the interactions between water, ecosystems and alterations of ecosystems by human influence. The objective of the research is clear and researched in a good manner. Furthermore, the methods are clearly written. The best method for water balance partitioning (runoff coefficient) is chosen. Sen and Altunkaynak (2006) investigated different versions of the runoff coefficient and thereby rainfall-runoff simulation and concluded that the runoff coefficient shows the least relative error compared to other methods. Besides, the choice of using different types of statistical methods increases the credibility of the results. In this manner strengths and weaknesses of all types of methods can be taken into account to come to a solid conclusion. Furthermore, using basins all around the world and the complete range of forest cover fractions increases the credibility and usage of the method worldwide. The figures are clearly presented and can directly be understood. The discussion section elaborates greatly on the processes that could be of importance in the relation between forest cover and water balance partitioning. Besides addressing the role of water- or energy-limited systems, also the role of forests is discussed. Because of the broad discussion, the mechanisms behind the results become very clear, which makes the final conclusion more convincing.

*Thank you for these comments.*

However, I am still not fully convinced by the conclusion that forest cover is an effective descriptor of basin attributes describing the long-term water balance for the high forest cover fraction (>0.5). Since 5 out of 14 basins fall outside the 95

*We used the LOESS regression as an exploratory tool to test for the presence of a statistically significant trend in the data. The results of the regression analysis (for which we used a non-parametric test) show that, overall, the relationship between forest cover and K is statistically significant at an alpha-level of 0.05. However, as illustrated in the comment, not all data points fall strictly on the regression line or its uncertainty bands (as expected). As we highlight in the paper, these regressions are not aimed to be predictive in nature, but rather they are used to test for general trends in the data. We appreciate the point and we will make sure to acknowledge that a portion of the basins do not fall within the confidence boundaries of the regression.*

Besides, the authors state that forest loss will force a system into a evaporation-dominated system. However, this is not well explained and other papers do not agree with this statement. Furthermore, the structure of the paper should be improved, in order to make the readability better. Some major revisions need thus to take place in

order to be able to accept the paper, which I will elaborate on in the next paragraphs.

*We appreciate the comment and expect to improve structure in a revised version of the manuscript. Based on our results, we propose the hypothesis that by losing forest cover, a basin can potentially move to an evaporation-dominated water balance partitioning. This is the case of, for example, basins in arid and semi-arid areas, where ET dominates the water-budget. However, we are aware that other papers do not agree with this statement (as discussed in P2L3-5). We discuss this by differentiating small vs. large scale (in both space and time) effects of forests on water balance partitioning, such that short-term effects of forest loss can lead to a general increase in streamflow while generally decreasing in the long-term in the absence of forests (leading to increased evaporation losses). Similarly, forest loss over large areas can affect atmospheric moisture transport processes that result in generally drier conditions, under which evaporation tends to dominate water balance partitioning. We use section 4.2 to describe a suite of mechanisms through which forest cover influences long-term water balance partitioning. However, and in response to RC1, in the revised version we will explicitly include an explanation of these mechanisms and how they relate to a potential increase in evaporation.*

**Major revisions**

The largest concern of this article is that the conclusion is not fully supported by the observations. The conclusion that forest is an effective descriptor of the basin attributes that are relevant for characterizing the long-term water balance in large basins of the world is not fully convincing, since the results (figure 2) show that 5 out of 14 basins (Vitim, Madeira, Mackenzie, Lena and Purus) with a high forest cover all outside the 95

*As mentioned previously, the regression analysis that we present is not predictive in nature, but rather it is used to explore the presence of a general pattern that relates forest cover to water balance partitioning. Our results show such pattern, and are supported by their statistical significance. We are aware that forest cover cannot be used as a sole*

*predictor of water balance partitioning, as hydrological regimes depend on other factors (including basins physical attributes, geological/geomorphological features, large-scale atmospheric circulation, among others). Therefore, we use our results to discuss potential mechanisms in which forest cover can affect hydrological partitioning and how those could be affected by forest loss, though examples of river basins in which the relationship is clearer. Discussing each basin's behavior is out of the scope, as this behavior can be influenced by each basin's particularities. However, in a revised version of the manuscript, we will explicitly acknowledge this point and highlight that even with statistically significant relationship between forest cover and hydrological partitioning, not all basins are expected to fall within the confidence bands of the specific model.*

A possible reason of the deviation could be found by Ellison et al. (2012), who evaluated the existing controversy about the influence of forest cover on the downstream water availability. They concluded through a review that the scale at which the influence of forest cover have been investigated matter. Forest cover influences the global precipitation by the evapotranspiration they produce. Removing forest has transboundary implications for local and global ET and the water regime. Demand-thinking looks only locally into the needs of a forest (water consumption) rather than larger scale at the creation of water supply to another area. Both processes are important and cannot be seen separately. Water supply through evapotranspiration raises precipitation in other regions as well, affecting the water balance partitioning in another catchment. This article put emphasis on the effect of forest cover loss on transboundary regions, which has not been investigated in this paper and could influence the investigated water balance partitioning in the basins.

*We thank this comment and appreciate the reference suggestion (which we included in our manuscript). One key point of our manuscript is to illustrate how long-term precipitation is partitioned between streamflow and evaporation, and conclude that increased forest cover leads to a relatively even partitioning of precipitation into these two fluxes. We acknowledge the role of continental evaporation in maintaining precipitation*

*regimes. We include basins in a region in which this process has been particularly highlighted (the Amazon). In this study, we have selected large basins in which precipitation recycling can be accounted for. In a revised version of the manuscript we will make sure to explicitly acknowledge the role of local-to-regional evaporation in continental moisture transport and its effects on continental precipitation.*

The lack of explanation of the basins deviating from the general pattern in figure 2 has a big impact on the credibility of the conclusion and should thus be explained well. From the paper it should become clear why these basins are showing different patterns regarding the water partitioning. If there is no clear reason, the conclusion for basins with a high forest fraction is not convincing, since 35

*As mentioned in a previous response, we are not proposing that forest cover as the only (or main) control of water balance partitioning. We acknowledge that other attributes are needed to provide a full explanation of a basin's behavior. However, we highlight the presence of a statistically-significant pattern and the potential implications of this, given that forest cover is highly vulnerable to global environmental change. We do not intend to explain each individual basin, as it is out of the scope of this paper, and we lack the tools to assess each basin.*

Besides, it is also concluded that the results provide insight in understanding and predicting potential consequences of forest cover loss on the continental water balance. It is addressed that forest loss can force a basin to go from P-halved to E-dominated, affecting the production of runoff and thus the river flow regimes. This conclusion is not fully supported as well. The first contradiction can be found in the discussion (line 23, page 11), stating that forest cover increase will not lead to increased evaporation and decreased runoff, but can be the other way around as well. This is in agreement with line 33, page 8 in the observations, that states that the relations between the partitioning patterns and forest cover are intended to be only descriptive and not predictive. Furthermore, Zhang et al. (2016) investigated the hydrological responses of forest cover change and to influence of spatial scale, climate, forest type and hydrological regime in large and small watersheds around the world. They achieved this by investigating the response intensity of annual runoff to forest cover change. Climate conditions were investigated through the Budyko Dryness Index in order to determine whether a watershed was energy or water-limited. Besides, they compared different forest cover types (broadleaf, mixed and coniferous) and hydrological regimes (rain and snow- dominated) to test for their significance. They concluded that an increase in annual runoff due to forest cover decrease is significant at several spatial scales. All in all, the discussion and observations and the paper thus concluded the opposite from the article's conclusion that a forest will change into an evaporation- dominated system. More explanation should be given in order to make the article's conclusion more convincing. It should become clear why these results give insight in the consequences of forest loss and why is will go into an evaporation-dominated system rather than runoff-dominated.

*We appreciate these observations, but do not necessarily agree in that they are contradictory. With these statements, we acknowledge that forest cover change does not always result in the same kind of streamflow-evaporation response but rather that the response is also affected by a number of other processes. In our previous responses, we have detailed our analytical rationale behind using a Budyko-like approach to classify hydrological partitioning regimes into energy- and water-limited. We include Zhang (2016) and the Budyko index in our manuscript. However, a more detailed discussion will be incorporated in the revised version.*

The last major concern is that spurious correlations could be present between the runoff coefficient and forest cover due to transboundary effects that are not taken into account in his paper. Although the runoff coefficient is shown to be the best method to partition the water balance (Sen and Altunkayak, 2006), this method does not take transboundary effects into account. Ellison et al. (2012) showed that these effects are relevant in forested basins. Other papers that put emphasis on this are e.g. Ellison et al. (2017) who reviewed a substantial amount of research on forest, water and

energy interactions and Debortoli et al. (2016) who investigated the relationship between forest cover and rainfall patterns in the Amazon Forest using rainfall timeseries between 1971 – 2010 and forest cover from LANDSAT5 satellite data. They found that on regional scale forest cover and seasonal rainfall are correlated. Furthermore, forest loss resulted in a decrease in evapotranspiration and important implications for rainfall thousands of kilometres downwind. Focussing on the runoff coefficient only evaluates the effect of either changes in precipitation, runoff or evaporation. It is possible due to transboundary effects that the results can be affected by more precipitation in the upstream parts of the catchment that also has a large forest cover (Ellison et al., 2017). Forest could thus influence the water balance partitioning of another basin as well, resulting in more complex relationships between forest loss and the runoff coefficient. Spurious correlation between forest cover and runoff should be excluded. This can be done by showing that the same results can be obtained using average evapotranspiration (precipitation minus runoff). Otherwise, the signal of the correlation found has not been identified clearly.

*We appreciate the suggestion to acknowledge regional scale effects of forest loss, particularly with respect to moisture transport and its potential effects on precipitation. We do not believe that these processes affect our results for multiple reasons: (1) Our precipitation data is based mostly on observations, which include (in case they occur) the effects of forest loss on recycled precipitations. (2) We include very large basins in the analysis, such that moisture divergence can be controlled for. (3) We use long-term statistics of hydrological partitioning and forest cover, such that inter-annual variability in both climate and forest cover can be accounted for. We are not completely sure about the use of "spurious correlation" in are results, as vegetation cover and runoff have been known to be related processes at the basin scale. We acknowledge that the effects of vegetation cover (particularly forests) on atmospheric moisture transport and continental dynamics need to be further studied, but believe that such study is out of the scope of this manuscript.*
Finally, some minor issues should be addressed.

**Minor Revisions**

The title should be more attractive. Now it is too general and does not include the main conclusion of the article. This could be changed into: "Forest cover as descriptor for long-term water balance partitioning". *We appreciate the suggestion and will consider refining the title in a revised version of the manuscript*

The structure of the article should be revised. In the discussion part 4.1 figure 4 is introduced, while in the methods they elaborate on the way how they obtained this data. In my opinion, this should be included in the observations and compared there, while the reason behind the possible connections should be discussed in the discussion. In the discussion session the observations should be discussed, while in the observation session no discussion should be started. Both parts of the paper should be clearly distinctive. Introducing new observations in the discussion is confusing, because one expects to already have received all information on the research done in this study. It is better to transfer the observations (line 10-18 and 26-30 of page 9) to the observational part of the paper and to transfer the challenges and discussion of the patterns showed in the observation part (line 24- 35 of page 8 and line 1-7 of page 9) to the discussion part, because this can be seen as recommendations for further research. However, what also could be done in order to be more clear, is to change the title of section 3 to something like: "Patterns in water balance partitioning" and section 4.1 should become 3.2, discussing the effect of water- and energy limitations. Then the discussion should only consist of the role of forests, bringing all observations together. However, the recommendations (line 24-35, page 8 and line 1-7, page 9) should always be discussed in the discussion section rather than the results. It is up to the authors to decide which method to use.

*We appreciate the suggested reorganization of the manuscript, which we will consider along with suggestions from the reviewers and other comments in the revised version*

*of the manuscript.*

Argument why a particular method is used. More emphasis should be given to the reason why the runoff coefficient rather than another method is chosen. Furthermore, it should explained why they used a 12-year average for the calculation of the k-value and not another timeframe. It is stated that forest cover is decreasing, it would also be interesting to investigate the changes in partitioning throughout the years and see what influence this change has on the water balance. Besides, a metric of forest is provided by constructing a global land cover map using the temporal mode for each pixel in 2001-2012 map series of MODIS-MCD12C1 with five land use classes. They constructed the land use maps in temporal mode, because landcover is not static but changing through time. This seems logic. However, since they used the runoff coefficient averaged of 12 years, it is not clear why to use temporal landcover data. Furthermore, they divided the land uses into five classes using the 16 classes of IGBP and only investigated forest or non-forested areas. They could also have chosen to investigate the influence of other land use types as well dividing this way, since they have a significant impact on the runoff generation (Mahe et al., 2004). With the division, they put all forested areas in one class, so including evergreen and deciduous. Explanation of the choices made will clarify these questions.

*The main focus of the manuscript is to identify general patterns in long-term water balance partitioning in relation to forest cover. The timescale of averaging water balance variables was selected based on data availability (a common period for all basins) such that is was large enough to control for year-to-year climate variability, which if not controlled for, could potentially affect the results. Since we used long-term averages of hydrological variables, we chose the most appropriate representation of vegetation cover for the same period. We decided to use the mode as the most representative description of vegetation cover. We agree that other vegetation cover types can have different effects on hydrological partitioning and that they need to be further investigated. However, including all vegetation types was out of the scope of this manuscript*

*that focused primarily on the role of forests. We acknowledge that different types of forests differ in their ecology and function, but we find the emergence of a common pattern from forests is particularly interesting, and are used to produce a general hypothesis about their role in hydrological partitioning.*

Ellison et al. (2017) should be included, because this paper reviews on the latest research done on the effect of forest cover on the water balance and vice versa. *We appreciate the suggestion, and will include the reference in the revised version of the manuscript*

Page 1, line 21: "For instance" can be deleted. *Corrected*

Page 2, line 1: change "for" to "in". *Corrected*

Page 2, line 1-2: "Panta Rhei – Everything flows" debate is not known by everyone and there should be elaborated on this. *We will expand context on the debate within the parenthetical note that refers to the supporting reference)*

Page 2, line 10-11, In the Amazon 40

Page 2, line 22: change "otherwise" in "the other way around". *Corrected*

Page 2, line 30: change "We choose to focus on" in "We focused on" *Corrected*

Page 3, line 3: delete "," in "precipitation, (P)" *Corrected*

Page 3, line 20-21: include the units of R, Q and A. *Corrected*

Page 3, line 27-30: provide the assumptions made using the snow-melt equivalent and the consequences of this. Expanded context will be provided in the revised version
Page 7: change "either" in "neither". *Corrected*

Page 9, line 30: delete "of". *Corrected*

Page 11, line 8: change "this" in "these". *Corrected*

Page 11, line 9: delete "the hypothesis". *Corrected*

Page 13, line 10: delete "." after "environments". *Corrected*

**References**

Bosilovich, M. G., Chern, J. D. (2006). Simulation of water sources and precipitation recycling for the MacKenzie, Mississippi, and Amazon River basins. Journal of Hydrometeorology, 7(3), 312-329.

Debortoli, N. S., Dubreuil, V., Hirota, M., Lindoso, D. P., Nabucet, J. (2017). Detecting deforestation impacts in Southern Amazonia rainfall using rain gauges. International Journal of Climatology, 37(6), 2889-2900.

Ellison, D., Morris, C. E., Locatelli, B., Sheil, D., Cohen, J., Murdiyarso, D., ... Gaveau, D. (2017). Trees, forests and water: Cool insights for a hot world. Global Environmental Change, 43, 51-61.

Ellison, D., N Futter, M., and Bishop, K.: On the forest cover–water yield debate: from demand-to supply-side thinking, Global Change Biology, 18, 806–820, 2012. Eltahir, E. A., Bras, R. L. (1994). Precipitation recycling in the Amazon basin. Quarterly Journal of the Royal Meteorological Society, 120(518), 861-880.

Mahe, G., Paturel, J. E., Servat, E., Conway, D., Dezetter, A. (2005). The impact of land use change on soil water holding capacity and river flow modelling in the Nakambe River, Burkina-Faso. Journal of Hydrology, 300(1), 33-43.

Sİğen, Z., Altunkaynak, A. (2006). A comparative fuzzy logic approach to runoff coefficient and runoff estimation. Hydrological Processes, 20(9), 1993-2009.

Zhang, M., Liu, N., Harper, R., Li, Q., Liu, K., Wei, X., Ning, D., Hou, Y., and Liu, S.: A global review on hydrological responses to forest change across multiple spatial scales: importance of scale, climate, forest type and hydrological regime, Journal of Hydrology, 2016.

550, 2017.